# SimpleGVR: A Simple Baseline for Latent-Cascaded Generative Video Super-Resolution

**Liangbin Xie**[1,2*] **Yu Li**[3] **Shian Du**[3] **Menghan Xia**[4†] **Xintao Wang**[5]

**Fanghua Yu**[2] **Ziyan Chen**[2] **Pengfei Wan**[5] **Jiantao Zhou**[1†] **Chao Dong**[2,6]

[1]State Key Laboratory of Internet of Things for Smart City, University of Macau
[2]Shenzhen Institutes of Advanced Technology, Chinese Academy of Sciences
[3]Tsinghua University   [4]HUST   [5]Kuaishou Technology   [6]Shenzhen University of Advanced Technology

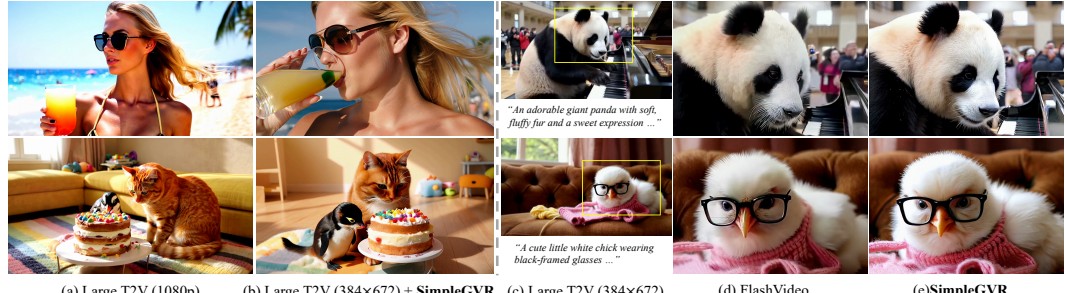

(a) Large T2V (1080p)   (b) Large T2V (384×672) + **SimpleGVR**   (c) Large T2V (384×672)   (d) FlashVideo   (e)**SimpleGVR**

Figure 1: Built upon the low-resolution latent outputs (e.g., $384 \times 672$ resolution) from the first-stage Large T2V model, SimpleGVR generates high-quality results that even surpass the 1080p outputs of the Large T2V model. Compared to FlashVideo, which also adopts a cascaded architecture, SimpleGVR produces more realistic and finer details.

## Abstract

Cascaded pipelines, which use a base text-to-video (T2V) model for low-resolution content and a video super-resolution (VSR) model for high-resolution details, are a prevailing strategy for efficient video synthesis. However, current works suffer from two key limitations: an inefficient pixel-space interface that introduces non-trivial computational overhead, and mismatched degradation strategies that compromise the visual quality of AIGC content. To address these issues, we introduce SimpleGVR, a lightweight VSR model designed to operate entirely within the latent space. Key to SimpleGVR are a latent upsampler for effective, detail-preserving conditioning of the high-resolution synthesis, and two degradation strategies (flow-based and model-guided) to ensure better alignment with the upstream T2V model. To further enhance the performance and practical applicability of SimpleGVR, we introduce a set of crucial training optimizations: a detail-aware timestep sampler, a suitable noise augmentation range, and an efficient interleaving temporal unit mechanism for long-video handling. Extensive experiments demonstrate the superiority of our framework over existing methods, with ablation studies confirming the efficacy of each design. Our work establishes a simple yet effective baseline for cascaded video super-resolution generation, offering practical insights to guide future advancements in efficient cascaded systems. Video visual comparisons are available here.

## 1 Introduction

Recent advancements (Chen et al., 2023b; Fridman et al., 2024; Voleti et al., 2022; Blattmann et al., 2023; Yang et al., 2024b; Polyak et al.; Ma et al., 2025; Kong et al., 2024; Seawead et al., 2025; Wan et al., 2025) in diffusion-based text-to-video (T2V) generation have markedly enhanced the visual

---

* Work done during an internship at KwaiVGI, Kuaishou Technology. † Corresponding authors

quality and coherence of synthesized videos. Leading models, such as Hunyuan (Kong et al., 2024) and Wan (Wan et al., 2025), rely on large DiT backbones with full self-attention to fuse spatial, temporal, and textual cues, producing coherent clips with rich detail. However, their computational cost grows quadratically with spatial resolution: directly generating 1080p video[1] in a single stage demands prohibitive computation and incurs long inference times.

To mitigate the substantial computational cost associated with generating high-resolution video, a prevailing and effective strategy is to adopt a cascaded generation pipeline, which first uses a powerful base T2V model to create a low-resolution resolution capturing the core semantic content and motion, followed by a lightweight video super-resolution (VSR) model to synthesize fine-grained details. Many works (Zhang et al., 2025a; Wang et al., 2025c; Zhang et al., 2025b) implement this strategy. However, we observe that these works treat the base and VSR models as two loosely-coupled components. They are merely a simple combination relying on an inefficient pixel-space interface: the base model's latent output is first decoded into a pixel-space video, upscaled via video-level interpolation, and subsequently re-encoded to serve as the input for the VSR model. These redundant VAE decoding and re-encoding steps introduce non-trivial computational overhead and increase inference time.

Beyond the architectural inefficiency, the performance of VSR models is limited by the mismatched degradation strategies. These models typically rely on either simple downsampling kernels (He et al., 2024) or more advanced two-stage degradation schemes (Chan et al., 2022b; Wang et al., 2025b; Zhang et al., 2025b). While the latter approach is effective for general video enhancement, it is ill-suited for AIGC content. A model trained with such a scheme tends to generate severe artifacts or compromise depth perception when applied to AIGC content from Large T2V models.

To address these two limitations, in this work, we propose SimpleGVR, a lightweight diffusion-based VSR model. It supports direct operating on the low-resolution latent representations produced by upstream T2V models, thereby eliminating redundant decoding and re-encoding steps. While channel concatenation is a more efficient conditioning strategy than alternatives like ControlNet (He et al., 2024) and token concatenation (Bai et al., 2025), the upsampling of the low-resolution (LR) latent via naive interpolation causes a loss of local detail. We therefore introduce a latent upsampler that preserves local structural integrity by first expanding the latent's channel and temporal dimensions, performing interpolation, and then reducing them. To address the second limitation of mismatched degradation strategies, we propose two methods designed to mimic the output characteristics of the upstream T2V model: (1) Flow-based degradation, where optical flow guides motion-aware color blending and adaptive blurring, and (2) Model-guided degradation, where noise is added to low-resolution video frames and partially denoised using the base T2V model. These strategies generate training pairs that better reflect the characteristics of base T2V model output.

Based on the architecture design and degradation strategies, we further optimize the training configuration of SimpleGVR in three key aspects. First, we propose a detail-aware timestep sampler that more effectively reconstructs high-frequency details compared to a uniform sampler. Second, we identify an optimal middle-range noise level (i.e., $0.3 \sim 0.6$) for the low-resolution input augmentation, which strikes a better balance between detail enhancement and structural correction. Finally, to enable practical application on long videos (e.g., 77 frames) under memory constraints, we introduce the interleaving temporal unit mechanism to extend the model from short to long sequences for both training and inference.

In summary, our main contributions are as follows: **1)** We present SimpleGVR, a lightweight diffusion-based VSR model that directly performs on the latent representations of large T2V models. Compared to using a single large T2V model to generate 1080p video end-to-end, the two-stage cascaded pipeline, which integrates SimpleGVR, achieves superior visual quality while reducing computational cost. **2)** We investigate different LR latent injection schemes and introduce a latent upsampler that effectively integrates information from low-resolution latents. **3)** We design two degradation schemes, namely flow-based degradation and model-guided degradation synthesis, to simulate the degradation characteristics of the base model's outputs. This ensures better alignment between the VSR model and its upstream generator. **4)** We present a set of training configurations, including a detail-aware sampler, a noise augmentation range, and the interleaving temporal unit, which improve the generative ability and practical applicability of SimpleGVR.

---

[1]Here 1080p roughly corresponds to a pixel area of about $1440^2$.

## 2 PRELIMINARY

Our work builds upon a pre-trained text-to-video foundation model that is composed of a 3D VAE (Kingma et al., 2013b), a T5 text encoder (Raffel et al., 2020), and a transformer-based latent diffusion module (DiT) (Chen et al., 2023a; Peebles & Xie, 2023). The DiT module processes latent representations using blocks of spatial self-attention, spatiotemporal attention, and text-guided cross-attention layers (conditioned on $c_{\text{text}}$), and we adopt the Rectified Flow framework (Esser et al., 2024) to define the linear path between the clean latent $z_0$ and its noisy counterpart $z_t$:

$$z_t = (1-t)z_0 + t\epsilon, \quad \epsilon \sim \mathcal{N}(0, I) \tag{1}$$

An ordinary differential equation (ODE) governs the denoising process, mapping $z_t$ back to $z_0$:

$$dz_t = v_\Theta(z_t, t, c_{\text{text}})dt, \tag{2}$$

where the velocity field $v$ is modeled by a neural network with parameters $v_\Theta$. During training, Conditional Flow Matching (CFM) (Lipman et al., 2022) is used to regress the velocity via the following objective:

$$\mathcal{L}_{\text{CFM}} = \mathbb{E}_{t, \epsilon \sim \mathcal{N}(0, I), z_0} \left[ \|(z_1 - z_0) - v_\Theta(z_t, t, c_{\text{text}})\|_2^2 \right]. \tag{3}$$

## 3 METHODOLOGY

Our cascaded video generation framework operates within a latent space defined by a pre-trained VAE. The framework comprises two core components: (i) A computationally intensive base Text-to-Video (T2V) model, which employs a Large DiT architecture to generate low-resolution video latent representations. (ii) A diffusion-based VSR model, termed SimpleGVR, which adopts a lightweight architecture to efficiently enhance the base model's output into high-resolution video latent representations and enhance the details. The overall framework structure is illustrated in Fig. 3. As the primary focus of this paper, our method addresses the task of the latter component.

In the following sections, we first present the overview of SimpleGVR (Sec. 3.1). Then, we investigate the low-resolution latent injection mechanism (Sec. 3.2). Subsequently, we describe the degradation simulation that generates training pairs (Sec. 3.3). Finally, we present three training configurations to further improve the performance and practical applicability of SimpleGVR (Sec. 3.4).

### 3.1 OVERVIEW OF SIMPLEGVR

Fig. 2 illustrates the training pipeline of SimpleGVR. To optimize SimpleGVR, we adopt a latent diffusion model. The high-resolution (HR) video $x_{\text{HR}} \in \mathbb{R}^{N_1 \times 3 \times H_1 \times W_1}$ and the corresponding low-resolution (LR) video $x_{\text{LR}} \in \mathbb{R}^{N_1 \times 3 \times H_2 \times W_2}$ are first encoded into latent representations via a 3D VAE (Kingma et al., 2013a), yielding HR $z_0 \in \mathbb{R}^{N_2 \times 8 \times H_3 \times W_3}$ and LR latents $c_0 \in \mathbb{R}^{N_2 \times 8 \times H_4 \times W_4}$. Here $N_1$ denotes the number of frames in a clip, while $H_1 \times W_1$ and $H_2 \times W_2$ are the spatial sizes of the high-resolution and low-resolution videos, roughly $1440^2$ and $512^2$, respectively. We omit the batch dimension for simplicity. The VAE we adopt downscales the spatio-temporal dimensions by $8 \times 8 \times 4$, which means $H_3 = H_1/8$, $H_4 = H_2/8$, and $N_2 = (N_1 - 1)/4 + 1$. To enable SimpleGVR to directly process the low-resolution latent produced by the Large T2V model during inference (shown in Fig. 3), unlike FlashVideo, the $c_0$ is derived from the original LR video rather than an upscaled LR video. Then, two independent random noises are then injected into both latents with different magnitudes, yielding noisy representations $z_t$ and $c_t$. $z_t$ refers to the noisy latent in the diffusion process, while $c_t$ denotes the noisy LR latent that serves as the conditioning input. To inject the information of $c_t$ into $z_t$, we explore this in the following section.

### 3.2 LOW-RESOLUTION LATENT INJECTION MECHANISM

As a lightweight VSR model, SimpleGVR is designed to enhance details while preserving the motion, structure, and content provided by the conditioning latent $c_t$. However, since $c_t$ and the model's

internal noisy latent $z_t$ have different dimensions, effectively incorporating this conditional guidance is a non-trivial design challenge. To address this, we explore several strategies for utilizing low-resolution latents and propose a low-resolution latent injection mechanism.

**Latent Interpolation + Channel Concatenation.** As illustrated in Fig. 2(b), since the spatial dimensions of $c_t$ and $z_t$ differ, the most straightforward approach is to do the bilinear interpolation for $c_t$ to match the dimensions of $z_t$, and then perform channel concatenation:

$$x_t = \text{patchify}\left([z_t, \text{bilinear}(c_t)]_{\text{channel-dim}}\right), \tag{4}$$

where $x_t \in \mathbb{R}^{L_1 \times D}$, $L_1 = (N_2 \times H_3 \times H_4)/4$, $D$ is the channel dimension.

**Token Concatenation + Self Attention + Token Drop.** Recent attempts (Tan et al., 2024; Bai et al., 2025) incorporate the condition input through token concatenation, thereby supporting condition inputs of arbitrary resolution. As shown in Fig. 2(c), $z_t$ and $c_t$ are patchified separately to obtain their respective tokens, which are then concatenated and fed into the self-attention module. Afterward, the tokens corresponding to $c_t$ are dropped, yielding $x_t$:

$$x_t = \text{drop}(\text{self\_attn}([\text{patchify}(z_t); \text{patchify}(c_t)])), \tag{5}$$

where $[\text{patchify}(z_t); \text{patchify}(c_t)]$ denotes the concatenation of patchified $z_t$ and $c_t$ tokens.

**Latent Upsampler + Channel Concatenation (ours).** To better preserve the layout and structural information of $c_t$, we propose a latent upsampler that enlarges $c_t$ to the same size as $z_t$ and then injects the upsampled information into $z_t$ via channel concatenation. Specifically, since $c_t$ contains compressed features produced by the encoder, we first expand its channel and temporal dimensions using two 3D residual blocks, followed by bilinear interpolation. We then employ another two residual blocks to reduce its temporal and channel dimensions back to match those of $z_t$.

$$x_t = \text{patchify}\left([z_t, \text{Res3D}(\text{Res3D}(\text{bilinear}(\text{Res3D}(\text{Res3D}(c_t)))))]_{\text{channel-dim}}\right), \tag{6}$$

where Res3D denotes the 3D residual block.

The architecture design of the latent upsampler is non-trivial. Its key component is the temporal expansion of the latent before spatial interpolation, which ensures that each frame in the expanded latent corresponds to a frame in RGB space. This design prevents inter-frame signal aliasing during the spatial upscaling process. To verify the importance of temporal expansion, we conduct another baseline: "3D ResBlocks + latent interpolation + channel concatenation" for comparison. Fig. 10 illustrates the comparative results.

**3D ResBlocks + Latent Interpolation + Channel Concatenation.** Similar to the latent upsampler, we apply two 3D ResBlocks to expand the low-resolution latent only along the channel dimension before interpolation. After interpolation, two additional 3D ResBlocks were used to reduce the latent dimensions, followed by channel concatenation with the high-resolution latent.

**Comparison and Discussion.** Our experiments show that incorporating $c_t$ into $z_t$ via the proposed latent upsampler combined with channel concatenation achieves better semantic fidelity and layout consistency in the final results, as illustrated in Fig. 10. Compared to latent interpolation, the latent upsampler first projects $c_t$ into a higher-dimensional space, enriching spatial and temporal details and enabling subsequent interpolation to more accurately blend structures and motion. In addition, channel concatenation preserves layout information more effectively than a single-layer token concatenation strategy.

Once trained, SimpleGVR can be directly applied to T2V generation pipeline, as shown in Fig. 3. Specifically, given a random low-resolution gaussian noise $c_T$, the large T2V model performs multiple denoising steps to produce a clean low-resolution latent $c_0$. This latent is then perturbed with a fixed level of random noise, and upsampled to yield $c$ with the latent upsampler. Concurrently, a high-resolution gaussian noise sample $z_T$ is randomly initialized. The noisy high-resolution latent $z_T$ and the conditioned latent $c$ are concatenated along the channel dimension and fed into the DiT blocks of SimpleGVR. Notably, the conditioning latent $c$ remains fixed throughout the denoising process. After the iterative refinement, the final clean high-resolution latent $z_0$ is decoded to obtain a high-quality 1080p video.

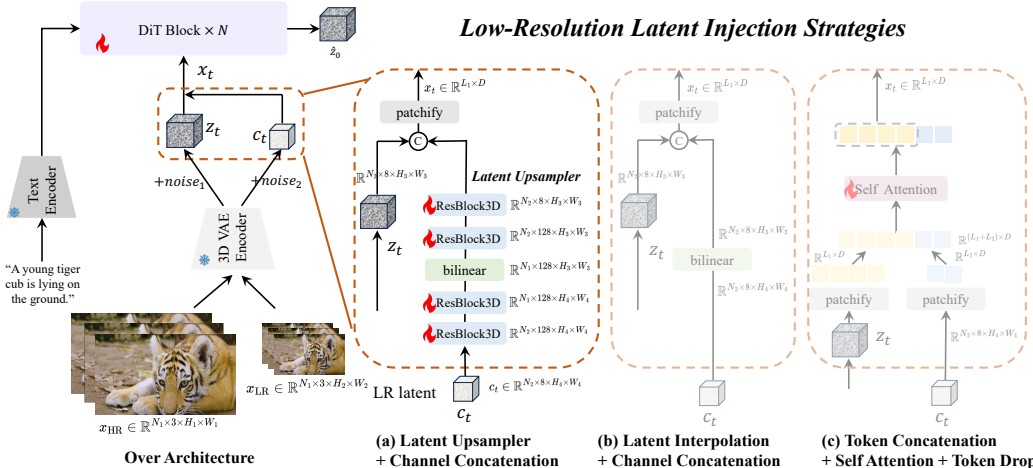

Figure 2: Overview of SimpleGVR. *Left:* The training pipeline of SimpleGVR. To eliminate redundant decoding and re-encoding steps during inference, the latent $c_t$ is not generated at the same spatial size as the high-resolution noisy latent $z_t$ from the very beginning. *Right:* Comparison of different low-resolution latent utilization strategies. (a) Latent upsampler and channel concatenation used in our paper; (b) Latent interpolation and channel concatenation; (c) Token concatenation, self-attention and token drop.

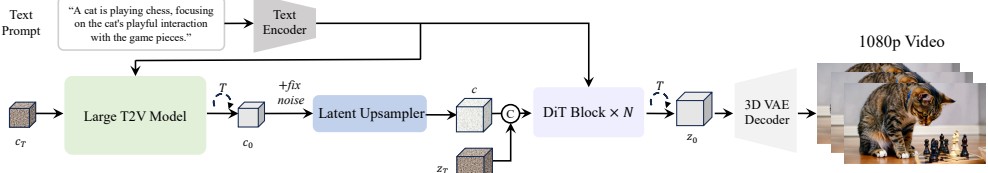

Figure 3: Cascaded high-resolution T2V pipeline. The large T2V model produces a low-resolution latent $c_0$, which is upsampled by the latent upsampler in SimpleGVR and concatenated with a randomly initialized $z_T$. The concatenated latent is iteratively denoised by DiT blocks to obtain $z_0$, which is then decoded into the final 1080p video.

## 3.3 DEGRADATION MODELING

### 3.3.1 FLOW-BASED DEGRADATION

Upon inspecting the base T2V outputs (see Fig. 4), we observe that unlike real-world low-quality videos, these video sequences do not exhibit severe degradations such as severe blur, noise, or compression. Instead, they primarily exhibit two entangled, motion-dependent characteristics: (1) frame-to-frame color blending (where hues from previous frames smear into the current one) and (2) localized motion blur. As conventional degradation models (Chan et al., 2022b) cannot replicate these effects, we design a flow-based degradation strategy to simulate these motion-dependent phenomena. The process is driven by the motion field, which is estimated between adjacent frames using the DIS optical flow algorithm (Kroeger et al., 2016).

This motion field is then leveraged to synthesize both distortions. To simulate color blending, we identify regions of significant movement and introduce randomized elliptical patterns to guide a color sampling process. Hues from corresponding locations in the previous frame are then blended into the current frame with a distance-based weighting, realistically mimicking the observed color smearing. For motion blur, the same motion field is used to generate adaptive, block-wise blur kernels. The parameters of each kernel (e.g., size and orientation) are determined by the local motion vectors. This ensures blur is only applied to moving regions and is aligned with the direction of motion, preserving the sharpness of static areas.

### 3.3.2 MODEL-GUIDED DEGRADATION

The primary objective of SimpleGVR is to learn a mapping from the output domain of large T2V models to high-quality video data. By constructing paired training samples where the low-resolution

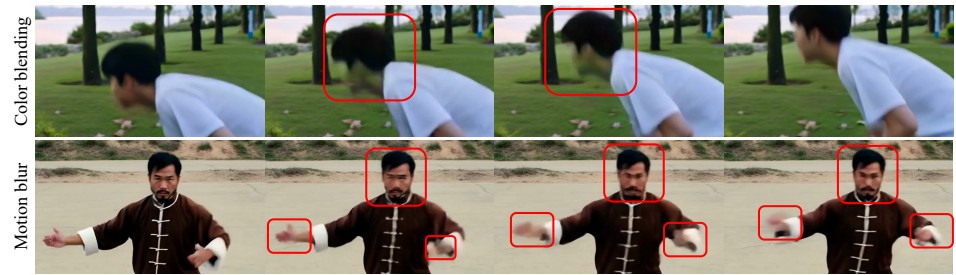

Figure 4: Visual artifacts in decoded videos from the Large T2V model. Dynamic regions exhibit noticeable local motion blur and color blending distortions.

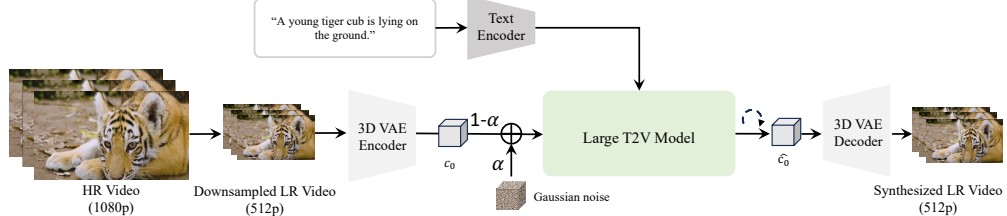

Figure 5: Model-guided degradation synthesis pipeline. The parameter $\alpha$ controls the strength of the added Gaussian noise, which also affects the structural alignment between $\hat{c}_0$ and $c_0$.

inputs are directly sourced from the T2V model outputs, SimpleGVR can be better aligned with the distribution and artifacts specific to the T2V model. Inspired by SDEdit (Meng et al., 2021), as shown in Fig. 5, we begin by downsampling a high-quality 1080p video to 512p and encoding it via a 3D VAE to obtain the latent $c_0$. This latent is blended with a gaussian noise under a predefined ratio $\alpha$, and the noisy latent is then partially denoised using the large T2V model to generate $\hat{c}_0$. A higher $\alpha$ pushes $\hat{c}_0$ closer to the T2V distribution but weakens its structural alignment with the original video. To balance realism and fidelity, we set $\alpha \in [0.3, 0.4]$, ensuring that $\hat{c}_0$ retains the overall layout of the source video while approximating the output domain of the Large T2V model.

## 3.4 TRAINING CONFIGURATION

To further enhance SimpleGVR's ability, we optimize the training configuration of SimpleGVR in three key aspects: the timestep sampling scheduling, the noise augmentation applied to the low-resolution (LR) branch, and the efficient training (i.e., interleaving temporal unit).

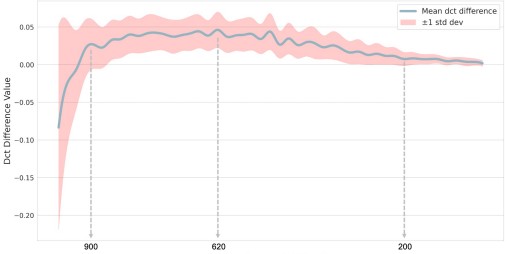

Figure 6: High-frequency variation curve over timesteps during inference.

Table 1: Quantitative comparison between the uniform sampler and the detail-aware sampler, demonstrating that the detail-aware sampler outperforms the uniform sampler in most metrics. These experiments are conducted on 17-frame inputs for 20K iterations.

| Sampler | MUSIQ | DOVER | | |
| --- | --- | --- | --- | --- |
| | | Technical | Aesthetic | Overall |
| Uniform | 62.04 | 18.58 | 98.78 | 68.94 |
| Detail-aware | 62.19 | 18.92 | 98.83 | 69.64 |

**Timestep Sampling Scheduling.** Since SimpleGVR focuses on detail synthesis, understanding which timesteps contribute most to enhancing visual details during denoising is crucial. To this end, we analyze high-frequency detail changes in the predicted $\hat{z}_t^0$ at each denoising step. Specifically, we sample 200 low-resolution 512p test videos and perform 50-steps inference using the SimpleGVR model trained with a uniform sampler. At each denoising timestep $t$, we obtain the latent $z_t$ and directly predict its corresponding clean signal $\hat{z}_t^0$. To quantify the high-frequency content of $\hat{z}_t^0$, we apply discrete cosine transform (DCT) and extract its high-frequency coefficients $\mathcal{H}(\hat{z}_t^0)$. We then compute the pairwise differences of these high-frequency components across timesteps to derive the detail variation curve shown in Fig. 6. The figure shows that detail gains primarily occur in the high and mid-noise regions, while the low-noise region contributes minimally. Based on this

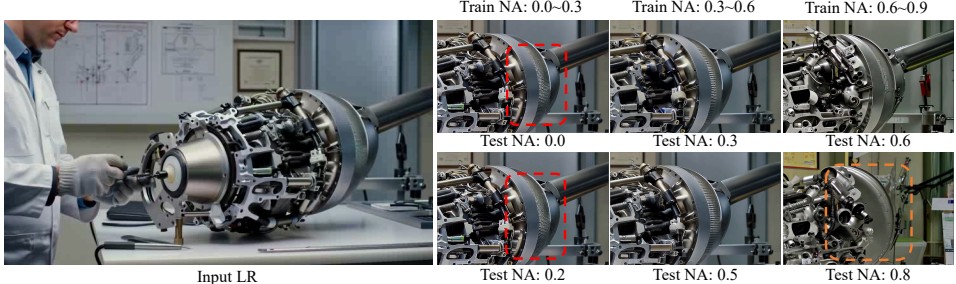

Figure 7: Visual results of SimpleGVR trained with different noise augmentation (NA) ranges.

observation, we propose a detail-aware sampler by normalizing this variation curve into a probability distribution. In the training phase, we derive different sampling probabilities for different time steps; in the inference phase, the sampling steps remain uniformly selected (e.g., 1000, 980, 960, ..., 0) as in standard diffusion processes. As demonstrated in Table 1, replacing the standard uniform sampler with our detail-aware version during training leads to improved performance.

**Noise Augmentation Effect.** The level of noise augmentation (NA) applied to the low-resolution latent is a critical hyperparameter, controlling the trade-off between structural fidelity to the input and the model's capacity for detail enhancement and correction. To identify a suitable range for SimpleGVR, we conduct experiments with three intervals: small $(0.0 \sim 0.3)$, middle $(0.3 \sim 0.6)$, and large $(0.6 \sim 0.9)$. As shown in Fig. 7, a large noise interval $(0.6 \sim 0.9)$ causes the model to disregard the input's global structure, leading to significant divergence in shape and color (highlight in orange box). Conversely, a small noise interval $(0.0 \sim 0.3)$ limits the model's generative capacity. When the input contains fine-grained structural errors (shown in the red box), the model becomes too faithful to this flawed input and fails to make corrections, preserving the messy details instead. Only the middle interval $(0.3 \sim 0.6)$ strikes an effective balance. It empowers the model to correct localized, fine-grained structural errors while remaining the overall content and global structure of the input video.

**Interleaving Temporal Unit.** Processing long video sequences (e.g., 77 frames) with full attention is often infeasible due to GPU memory constraints. We address this by first training SimpleGVR on short 17-frame clips and then extending its capabilities using our interleaving temporal unit mechanism. As illustrated in Fig. 8, the long latent sequence is divided into smaller, efficient windows along the temporal dimension when fed into the transformer blocks $l$ (where $l$ denotes the entire sequence of transformer blocks). In even-numbered blocks $l_{2k}$ and $l_{2k+2}$, the sequence is partitioned into four

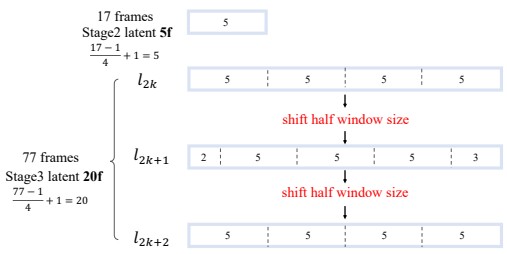

Figure 8: Visualization of the interleaving temporal unit mechanism.

non-overlapping windows. In odd-numbered blocks $l_{2k+1}$, to enable information exchange across windows, the attention windows are shifted by half of their size along the temporal axis, following a Swin-style Liu et al. (2021) attention mechanism. This alternating scheme allows the model to perform efficient temporal attention while maintaining long-range dependencies.

# 4 EXPERIMENTS

## 4.1 IMPLEMENTATION DETAILS

### 4.1.1 TRAINING DATASET

We design an automated filtering pipeline and collect approximately 840K high-quality video clips (each contains more than 77 frames) from the Internet to construct our training dataset. Specifically, we first discard videos that are overly bright or dark. Then, for each video, we uniformly sample 10 frames and compute two metrics: the average MUSIQ score (Ke et al., 2021) and the Laplacian variance, which reflects the level of spatial detail or sharpness. Videos with an average MUSIQ score below 40 or a Laplacian variance below 30 are discarded.

Table 2: Quantitative comparison on AIGC100 dataset. **Bold** and underline indicate the best and second best performance.

| Method | MUSIQ | CLIPIQA | MANIQA | NIQE(↓) | $E^*_{warp} \times 10^3$(↓) | DOVER | | | VBench Metrics | | | | | |
| | | | | | | Technical | Aesthetic | Overall | Background Consistency | Subject Consistency | Aesthetic Quality | Imaging Quality | Motion Smoothness | Average Score |
|---|---|---|---|---|---|---|---|---|---|---|---|---|---|---|
| RealBasicVSR | 57.55 | 0.5970 | 0.4591 | 4.6062 | 4.485 | 12.27 | 98.66 | 61.84 | 93.73 | 93.98 | 61.63 | 72.76 | 98.70 | 84.16 |
| VEnhancer | 40.03 | 0.5034 | 0.3429 | 5.3319 | 2.796 | 15.38 | 98.32 | 62.54 | 94.59 | 94.44 | 59.98 | 64.22 | 99.16 | 82.48 |
| Upscale-A-Video | 36.35 | 0.4744 | 0.3033 | 5.7165 | 4.314 | 12.43 | 98.29 | 59.04 | 95.96 | 94.41 | 61.26 | 63.85 | 98.99 | 82.89 |
| STAR | 46.73 | 0.5469 | 0.3743 | 4.9787 | **2.409** | 18.17 | 98.66 | 67.76 | 96.17 | 94.43 | 62.24 | 67.24 | 99.01 | 83.82 |
| Flashvideo | 53.71 | 0.5818 | 0.4262 | 4.8130 | 4.314 | 17.51 | 98.61 | 67.35 | 96.14 | 95.14 | 61.94 | 68.04 | 98.72 | 84.00 |
| SeedVR (7B) | 56.77 | 0.6176 | 0.4328 | 4.3025 | 3.800 | 18.05 | 97.40 | 61.87 | 94.80 | 93.80 | 63.82 | 69.49 | 98.51 | 84.08 |
| SeedVR2 (7B) | 53.51 | 0.6179 | 0.4242 | 4.3552 | 3.814 | 17.71 | 97.51 | 61.88 | 94.84 | 93.80 | 63.63 | 69.42 | 98.55 | 84.05 |
| DOVE | 60.34 | 0.5982 | 0.4332 | 4.7323 | 3.180 | 16.81 | 97.63 | 61.54 | 96.89 | 94.02 | 62.98 | 69.17 | 98.87 | 84.39 |
| MGLD | 52.19 | 0.6142 | 0.4260 | 4.1880 | 3.877 | 12.62 | 97.59 | 56.97 | 96.21 | 94.61 | 61.57 | 70.96 | 98.67 | 84.40 |
| DLoRAL | 58.57 | 0.5975 | 0.4302 | 4.5683 | 3.704 | 14.23 | 97.61 | 58.36 | 95.84 | 94.07 | 64.21 | 69.20 | 98.63 | 84.39 |
| Ours | **62.35** | **0.6768** | **0.4956** | **4.1665** | 2.592 | **20.44** | **98.88** | **71.34** | 95.35 | 94.32 | 62.84 | 71.91 | 98.74 | **84.63** |

### 4.1.2 TESTING DATASET

Based on this output, we collect a dataset, AIGC100, which contains 100 video clips. To ensure the diversity of the test set, this dataset covers a wide range of scenarios, including different subjects (e.g., humans and animals), various camera motions, and diverse backgrounds. More details of the AIGC100 dataset can be found in the supplementary material. In addition, to make more comprehensive evaluation, we construct VBench110 by randomly selecting 10 prompts from each of VBench's (Huang et al., 2024) 11 categories. The area of each clip is approximately $512^2$.

### 4.1.3 METRICS

Since there is no ground-truth reference for the AIGC100 and VBench110 datasets, we adopt several no-reference metrics to evaluate both frame-level and video-level quality. Specifically, we employ MUSIQ (Ke et al., 2021), MANIQA Yang et al. (2022), CLIPIQA Wang et al. (2023) for single-frame perceptual quality, DOVER (Wu et al., 2023) for overall video quality, and a suite of metrics from VBench (Huang et al., 2024) that assess various aspects of AIGC videos, including background consistency, subject consistency, aesthetic quality, imaging quality, and motion smoothness. Meanwhile, we adopt the flow warping error $E^*_{warp}$ Lai et al. (2018), to assess temporal consistency.

### 4.1.4 TRAINING DETAILS

SimpleGVR is trained on 16 GPUs with a total batch size of 32. We use the AdamW optimizer (Loshchilov & Hutter, 2017) with a learning rate of $5 \times 10^{-5}$, and randomly replace the text prompt with a null prompt in 10% of cases to enhance robustness. The training pipeline is divided into three stages. In the first stage, initialized from a pretrained 1B T2V model, SimpleGVR is trained for 20K iterations on 17-frame inputs using training pairs constructed via the degradation process in RealBasicVSR (Chan et al., 2022b). In the second stage, we fine-tune the model for an additional 10K iterations on a dataset (30K) generated using the proposed degradation strategies. In the third stage, based on the dataset synthesized in the previous two stages, we continue fine-tuning SimpleGVR with 5K iterations and extend the temporal range to 77 frames by using the interleaving temporal unit mechanism. During the whole training pipeline, we adopt the proposed detail-aware sampler, and the LR branch is injected with noise sampled from the range [0.3, 0.6].

## 4.2 COMPARISON WITH SOTA METHODS

We compare SimpleGVR with existing state-of-the-art methods, RealBasicVSR (Chan et al., 2022b), Upscale-A-Video (Zhou et al., 2024), VEnhancer (He et al., 2024), MGLD, STAR (Xie et al., 2025), SeedVR (Wang et al., 2025b), DiffVSR Li et al. (2025), MGLD Yang et al. (2024a), DOVE Chen et al. (2025), DLoRAL Sun et al. (2025), SeedVR2 Wang et al. (2025a), FlashVideo (Zhang et al., 2025b). For fair comparison, we set the inference steps of FlashVideo to 50. As shown in Table 2, SimpleGVR achieves the best performance on MUSIQ, MANIQA, CLIPIQA, and DOVER. Moreover, regarding the comprehensive metrics proposed in VBench, SimpleGVR also achieves the highest average score. For the temporal consistency metric $E^*_{warp}$, SimpleGVR also achieves competitive performance. Qualitative comparisons are presented in Fig. 9. Compared to other methods, for human faces, SimpleGVR produces finer and more realistic details. In contrast, other methods either struggle to generate sufficient detail or create noticeable artifacts. More visual comparisons can be found in the appendix and here.

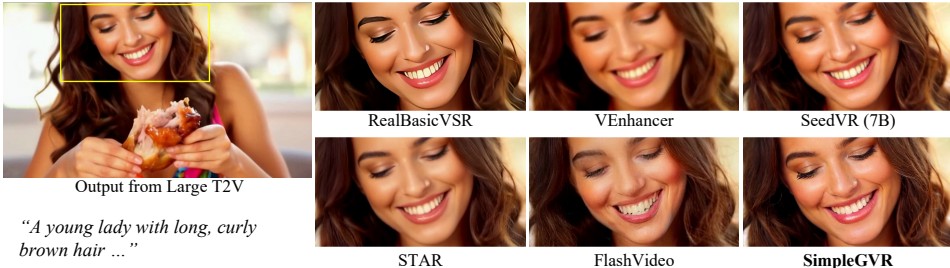

Figure 9: Qualitative comparison on AIGC100 dataset. Our SimpleGVR is capable of generating more realistic details than our methods. More visual comparisons can be seen in the appendix.

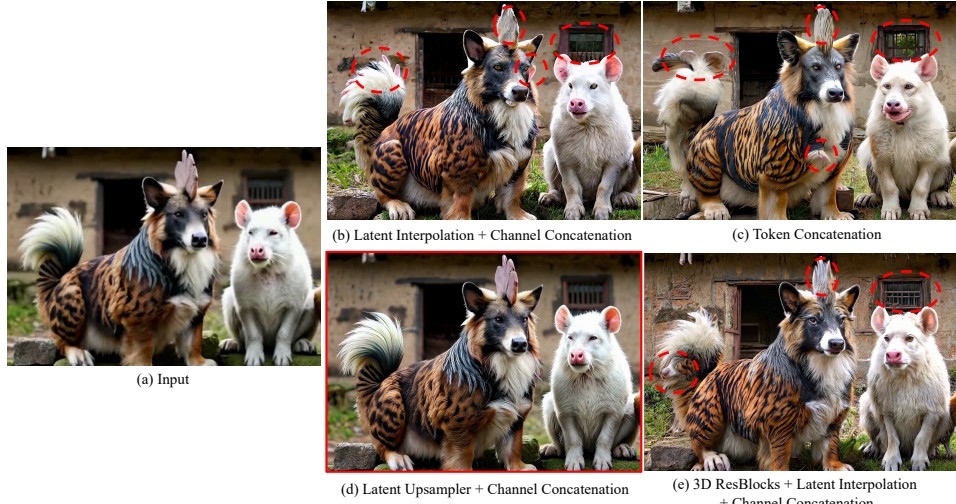

Figure 10: Ablation on low-resolution latent injection techniques. The proposed "latent upsampler + channel concatenation" can better preserve the layout and semantic content of the original input.

## 4.3 ABLATION STUDY

### 4.3.1 ABLATION ON LOW-RESOLUTION LATENT INJECTION TECHNIQUES

To verify the effectiveness of our "latent upsampler + channel concatenation", we compare our "latent upsampler + channel concatenation" strategy with three alternatives: "latent interpolation + channel concatenation", "token concatenation + self-attention + token drop", "3D ResBlocks + latent interpolation + channel concatenation". Both qualitative and quantitative comparisons, as presented in Fig. 10 and Tab. 8, clearly demonstrate that our low-resolution latent injection technique achieves better performance. As shown in Fig. 10 (highlighted with red circles), compared with our "latent upsampler + channel concatenation" approach, the other three alternative methods show less ability to preserve the original input layout and may slightly deviate from its semantic content (e.g., an extra ear appears in Fig 10 (b), while unnatural artifacts on the tail in Fig 10 (c, e)). Although the variant "3D ResBlocks + latent interpolation + channel concatenation" also employs additional 3D ResBlocks, performing only channel expansion does not sufficiently preserve the layout and semantic content of the latent.

### 4.3.2 EFFECTIVENESS OF DEGRADATION STRATEGIES

To ensure better alignment between the VSR model and its upstream generator, we propose two degradation strategies. As shown in Tab. 4, starting from a first-stage model trained on training pairs constructed with the degradation strategy of RealBasicVSR, we progressively incorporate data synthesized with our proposed degradation strategies into the training set for further training. The experimental results demonstrate the effectiveness of these two degradation strategies.

Table 3: Quantitative comparison between different low-resolution latent injection techniques. These experiments are conducted under the setting of 17 input frames for comparison.

| LR Feature Utlization | MUSIQ | DOVER | | | Vbench | | | | | |
|---|---|---|---|---|---|---|---|---|---|---|
| | | Technical | Aesthetic | Overall | Background Consistency | Subject Consistency | Aesthetic Quality | Imaging Quality | Motion Smoothness | Average Score |
| Interpolation + Channel Concatenation | 60.23 | 15.75 | 97.31 | 59.34 | 96.45 | 96.96 | 61.25 | 72.72 | 98.20 | 85.12 |
| Token Concatenation | 60.31 | 15.63 | 97.14 | 58.03 | 96.56 | 96.85 | 57.27 | 71.81 | 98.86 | 84.27 |
| **Upsampler + Channel Concatenation (ours)** | **62.06** | **16.25** | **97.60** | **61.25** | 96.64 | 97.02 | 61.49 | 72.79 | 98.36 | **85.26** |
| 3D ResBlocks + Latent Interpolation + Channel Concatenation | 61.75 | 15.18 | 97.50 | 59.43 | 96.43 | 96.96 | 60.51 | 71.89 | 98.27 | 84.81 |

Table 4: The effectiveness of our proposed degradation strategies. From top to bottom, we progressively add the synthesized paired dataset as part of the training set. These experiments are conducted under the setting of 17 input frames for comparison.

| Degradation Settings | MUSIQ | DOVER | | | Vbench | | | | | |
|---|---|---|---|---|---|---|---|---|---|---|
| | | Technical | Aesthetic | Overall | Background Consistency | Subject Consistency | Aesthetic Quality | Imaging Quality | Motion Smoothness | Average Score |
| Degradation(Chan et al., 2022b) | 62.06 | 16.25 | 97.60 | 61.25 | 96.64 | 97.02 | 61.49 | 72.79 | 98.36 | 85.26 |
| + Flow-based Degradation | 61.89 | 18.21 | 97.72 | 63.41 | 96.57 | 97.03 | 61.70 | 73.11 | 98.39 | 85.36 |
| + Model-guided Degradation | **62.19** | **18.92** | **98.83** | **69.64** | 96.95 | 96.96 | 62.59 | 73.37 | 98.78 | **85.73** |

## 4.4 COMPARISON IN T2V: END-TO-END VS. CASCADED

We also compare the performance of two different T2V paradigms: a large T2V model that directly generates 1080p videos (i.e., end-to-end), versus a large T2V model that first produces 512p latent representations followed by the SimpleGVR module to generate 1080p outputs (i.e., cascaded). As shown in Tab. 5, the cascaded paradigm achieves better performance on quality metrics than the end-to-end paradigm. On other metrics that measure diverse aspects of videos (i.e., smoothness and consistency), the results under both paradigms are comparable. The visual comparison is here.

To validate that the cascaded approach reduces computational overhead, we measure the generation time for each paradigm. As our proposed low-resolution latent injection mechanism removes the intermediate decoding and re-encoding steps, we focus the comparison on the core DiT processing time. The comparison reveals a substantial efficiency gain, with generation times of 950s for the end-to-end paradigm versus 283s for our approach. We note this result is achieved using 50 inference steps, a number with considerable redundancy that we aim to optimize in future work.

Table 5: Quantitative comparison between two different T2V paradigms on AIGC100 dataset.

| Method | MUSIQ | DOVER | | | Vbench | | | | | | Inference time (s) |
|---|---|---|---|---|---|---|---|---|---|---|---|
| | | Technical | Aesthetic | Overall | Background Consistency | Subject Consistency | Aesthetic Quality | Imaging Quality | Motion Smoothness | Average Score | |
| End-to-End | 56.77 | 18.82 | 97.27 | 62.32 | 96.04 | 95.16 | 63.45 | 67.69 | 98.89 | 84.25 | 950 |
| Cascaded | **62.35** | **20.44** | **98.88** | **71.34** | 95.35 | 94.32 | 62.84 | 71.91 | 98.74 | **84.63** | **283** |

## 5 CONCLUSIONS

In this work, we propose SimpleGVR, a lightweight video super-resolution model that operates entirely in the latent space to eliminate the redundant decoding and re-encoding steps. To enable effective conditioning within this latent-space framework, we introduce a latent upsampler for detail-preserving injection of low-resolution information. We further align SimpleGVR with its base generator through two AIGC-centric degradation strategies for synthesizing training pairs. Then SimpleGVR is optimized by a suite of training configurations, including a detail-aware sampler, a suitable noise augmentation range and an efficient long-video mechanism, which enhance both generative quality and practical applicability. Experimental results demonstrate the superiority of SimpleGVR, providing an effective baseline for future research in cascaded video synthesis.

## ACKNOWLEDGMENT

This work was supported in part by Macau Science and Technology Development Fund under 001/2024/SKL, 0119/2024/RIB2, 0110/2025/R1B2, and 0022/2022/A1; in part by Research Committee at University of Macau under MYRG-CRG2025-00031-FST and MYRG-GRG2025-00086-FST; in part by the Guangdong Basic and Applied Basic Research Foundation under Grant 2024A1515012536; in part by RGC General Research Fund No. 12200725; in part by the National Natural Science Foundation of China (Grant No. 62276251, 62502169).

**Ethics statement.** This work includes a user study involving human subjects. All participants were informed of the study's purpose and provided consent prior to participation. The study design and procedures were conducted in a manner consistent with ethical standards to ensure the protection of participants' rights and privacy. In addition, as with any generative model, our method carries the risk of potential misuse. We emphasize that the system should be applied responsibly and urge caution to avoid malicious or harmful applications.

**Reproducibility Statement.** To ensure the reproducibility of our work, we will ensure the following points. **Code:** Our code and model will be made publicly available, including necessary scripts. **Data:** Detailed descriptions of our data processing are provided in Sec. 4.1. **Experimental Setup:** We have stated all experimental configurations, including hyperparameters, hardware specifications in the Implementation Details of the main paper. **Model Architecture:** The architecture details are described in method part.

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

## A  APPENDIX

In this appendix, we include the following:

- Related work.
- Effectiveness of degradation strategies.
- Quantitative comparisons on the VBench110 dataset.
- User study.
- More visual comparisons.
- Details of the AIGC100 dataset.
- Performance on real-world low-quality videos.
- Performance on the output of other base T2V models (Wan (Wan et al., 2025), CogVideoX (Yang et al., 2024b)).
- Limitation.
- Discussions.

### A.1  RELATED WORK

**Cascade Diffusion Models.**  Cascade architectures have been widely explored in text-to-image, text-to-video and image-to-video generation (Li et al., 2023b; Saharia et al., 2022; Zhang et al., 2025b; 2023), where multi-stage designs are employed to address the challenge of generating high-resolution outputs. Typically, a low-resolution sample is first generated, followed by a specialized model to progressively refine details. Among these, FlashVideo (Zhang et al., 2025b) is most related to our work. It begins second-stage generation from low-quality video inputs rather than pure guassian noise, enabling efficient high-resolution synthesis with only 4 function evaluations. However, SimpleGVR differs in two key aspects. First, we treat the low-resolution latent as a condition rather than directly using it as the input, allowing the model not only to leverage the coarse content contained in the low-resolution latent but also to flexibly correct its structural errors. Second, we introduce two degradation strategies that explicitly simulate the characteristic degradations from the first-stage T2V generator.

**Degradation Models in Restoration.**  Degradation modeling is important for effective image and video restoration. Traditional models (Dong et al., 2014; 2016; Gu et al., 2019) often rely on simple assumptions like bicubic downsampling or gaussian blur, which fail to capture complex real-world degradations. Prior works such as BSRGAN (Zhang et al., 2021), Real-ESRGAN (Wang et al., 2021), and video-oriented methods like RealBasicVSR (Chan et al., 2022b) simulate more complicated degradations, including blur, noise, and compression, improving robustness on real-world low quality images and videos. However, these models are designed for real-world scenarios and do not account for the unique distortions in AIGC-generated videos, such as motion blur and color blending. These AIGC-specific artifacts require specialized degradation modeling. To this end, we propose two degradation strategies: a flow-based degradation scheme and a model-guided degradation scheme via SDEdit. Together, they enable the generation of training pairs that better mimic the output characteristics of the first-stage T2V generator.

**Video Restoration.**  Early video restoration (VR) methods (Chan et al., 2021; 2022a; Chen et al., 2024; Li et al., 2023a; Liang et al., 2022; Wang et al., 2019; Youk et al., 2024; Xu et al., 2025) rely on synthetic data, limiting real-world performance. Later works (Chan et al., 2022b; Xie et al., 2023; Zhang & Yao, 2024) shift toward real scenarios but still struggle with texture realism. Diffusion-based approaches (He et al., 2024; Wang et al., 2025c; Li et al., 2025; Wang et al., 2025b; Zhang et al., 2025b; Rota et al., 2024) leverage generative priors to achieve more realistic and coherent video restoration. However, all these methods require decoded RGB frames and cannot operate directly on latent representations, making them less suitable for T2V pipelines. In contrast, our SimpleGVR performs upsampling and refinement directly in the latent space of the upstream generator, enabling seamless integration with generative video models.

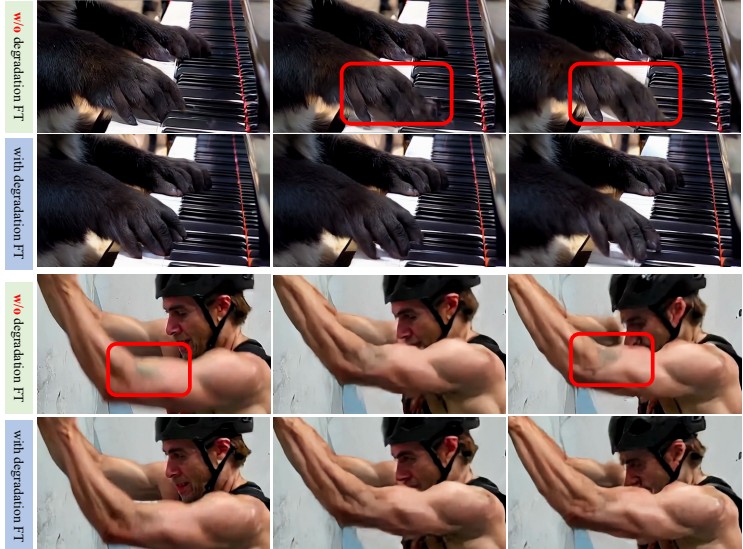

Figure 11: Visualization of three consecutive frames generated by SimpleGVR. "w/o degradation FT" indicates that SimpleGVR is trained only with the conventional degradation(Chan et al., 2022b), without fine-tuning using the proposed degradation strategies.

## A.2 EFFECTIVENESS OF DEGRADATION STRATEGIES

In Fig. 11, we demonstrate the effectiveness of the proposed degradation strategies. As shown in the first and third rows, SimpleGVR trained solely with the conventional degradation(Chan et al., 2022b) exhibits noticeable temporal inconsistencies in motion areas, such as the panda's paw, and suffers from color blending artifacts, particularly evident in the human arm. After fine-tuning with training pairs generated by the two proposed degradation schemes, SimpleGVR effectively mitigates abrupt changes in motion regions across consecutive frames and can eliminate color blending distortions.

## A.3 MORE QUANTITATIVE COMPARISONS

As we mentioned in the main paper, to make a more comprehensive evaluation, we also construct a dataset, VBench110, by randomly selecting 10 prompts from each of VBench's (Huang et al., 2024) 11 categories. The quantitative comparison of different methods on this dataset is shown in Tab. 6. It can be observed that SeedVR achieves the best performance on the MUSIQ metric. For other metrics, such as DOVER and the VBench average score, SimpleGVR attains the best results.

Table 6: Quantitative comparison on VBench110 dataset. **Bold** and underline indicate the best and second best performance.

| Method | MUSIQ | DOVER | | | Vbench | | | | | |
| | | Technical | Aesthetic | Overall | Background Consistency | Subject Consistency | Aesthetic Quality | Imaging Quality | Motion Smoothness | Average Score |
|---|---|---|---|---|---|---|---|---|---|---|
| RealBasicVSR | 57.22 | 11.30 | 98.41 | 58.87 | 94.55 | 95.26 | 64.79 | 72.71 | 98.96 | 85.25 |
| VEnhancer | 48.07 | 14.29 | 98.35 | 62.14 | 95.14 | 95.43 | 63.71 | 68.19 | 99.34 | 84.36 |
| Upscale-A-Video | 38.90 | 10.13 | 98.16 | 55.23 | 96.88 | 95.65 | 64.46 | 66.90 | 99.28 | 84.63 |
| STAR | 53.07 | 15.97 | 98.51 | 64.95 | 96.44 | 95.54 | 64.72 | 70.62 | 99.17 | 85.30 |
| Flashvideo | 54.31 | 14.46 | 98.35 | 62.40 | 96.06 | 95.10 | 62.48 | 69.64 | 98.96 | 84.45 |
| SeedVR (7B) | **61.64** | 15.14 | 97.14 | 58.18 | 95.57 | 95.23 | 66.04 | 71.18 | 98.71 | 85.35 |
| Ours | 60.20 | **16.50** | **98.53** | **65.71** | 95.86 | 95.34 | 65.05 | 71.98 | 99.02 | **85.45** |

## A.4 USER STUDY

Considering that existing IQA metrics (Ke et al., 2021; Huang et al., 2024; Wu et al., 2023) cannot fully capture perceptual performance, we further conduct a user study. Specifically, we randomly select 20 low-resolution clips from AIGC100 and VBench110. For each clip, we obtain the results of RealBasicVSR (Chan et al., 2022b), VEnhancer (He et al., 2024), STAR (Xie et al., 2025),

FlashVideo (Zhang et al., 2025b), SeedVR (Wang et al., 2025b) and SimpleGVR, and randomly shuffle the order of these method results. For each set of clips, we ask participants to independently select the two best videos. The first one is the best video with the highest visual quality (i.e., Detail Quality), and the second one is the video with the best temporal consistency or lowest temporal flickering (i.e., Temporal Consistency). The result is shown in Fig. 12. We can find that over 50% of participants prefer the details generated by SimpleGVR. Meanwhile, more than 43% of participants consider that SimpleGVR provides better temporal consistency. This further suggests that SimpleGVR is superior to other methods.

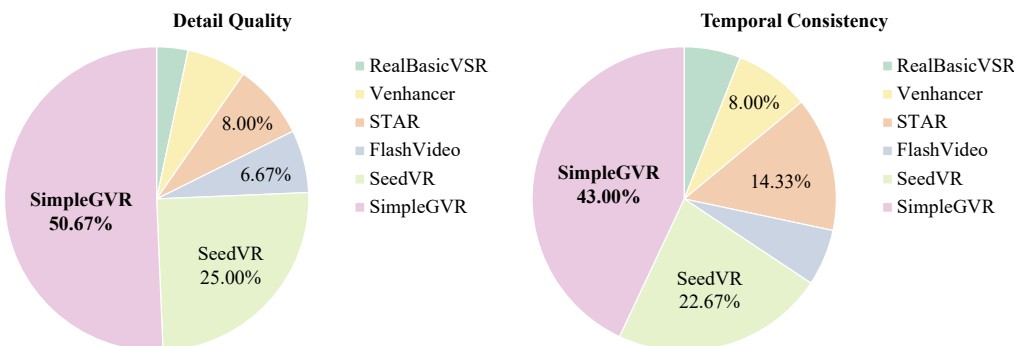

Figure 12: The results of user studies, comparing the results generated by RealBasicVSR, VEnhancer, STAR, FlashVideo, SeedVR, and SimpleGVR.

To further compare SimpleGVR with more recent or representative approache (i.e., MGLD Yang et al. (2024a), DiffVSR Li et al. (2025), DOVE Chen et al. (2025), DLoRAL Sun et al. (2025), and SeedVR2 Wang et al. (2025a)), we conduct an additional user study. This study follows a setting similar to the one described above. As shown in Fig. 13, the outputs of SimpleGVR are consistently preferred over those of the other methods.

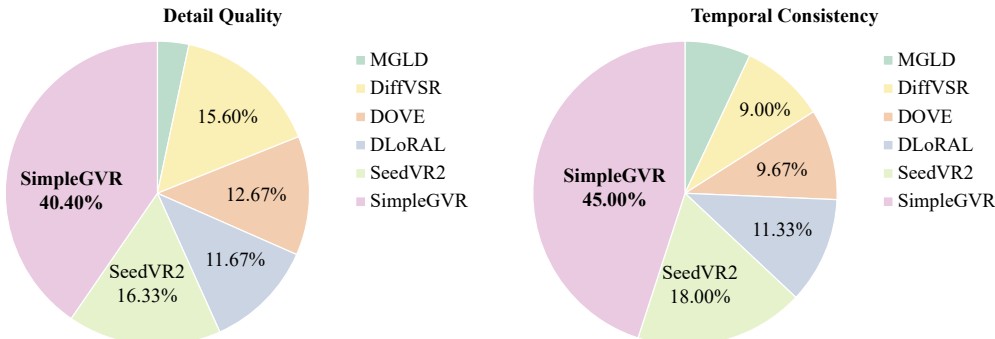

Figure 13: The results of user studies, comparing the results generated by MGLD, DiffVSR, DOVE, DLoRAL, SeedVR2, and SimpleGVR.

## A.5 MORE VISUAL COMPARISONS

Fig. 14 presents additional visual comparisons between SimpleGVR and other VSR methods. From the first group of results, it can be observed that, compared with other approaches, SimpleGVR exhibits stronger generative capability, successfully restoring the previously distorted guitar string from the upstream T2V model to a perfectly straight state. Besides, for animal fur, SimpleGVR is also able to generate richer and finer details. Please refer here for a better visualization.

## A.6 DETAILS OF THE AIGC100 DATASET.

AIGC100 consists of 100 low-resolution videos generated by the base T2V model. As shown in Fig. 15, we present the category composition and motion-intensity distribution of the AIGC100

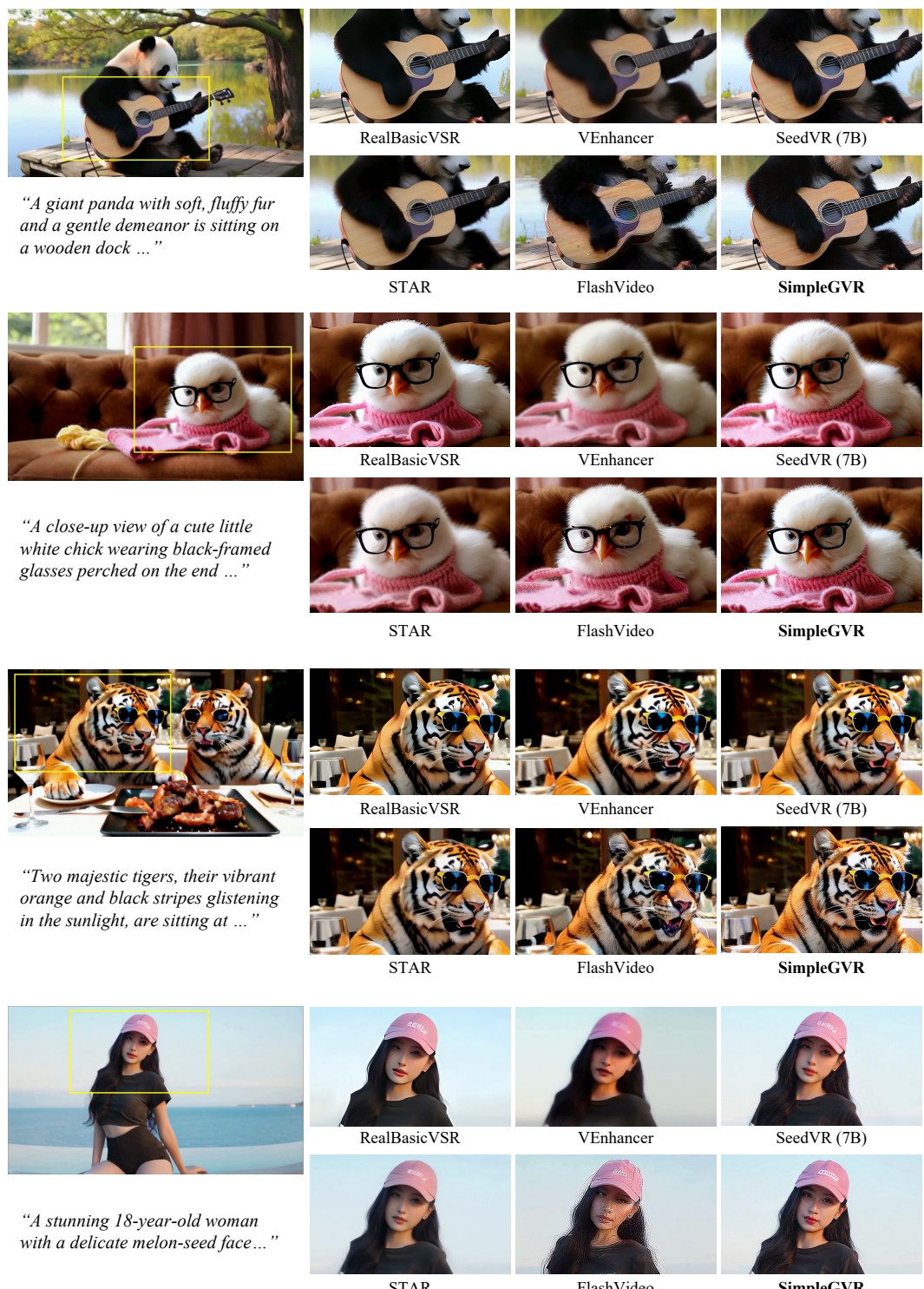

Figure 14: Qualitative comparisons with state-of-the-art methods. Our SimpleGVR is capable of generating more realistic details than our methods.

dataset. It encompasses a wide variety of semantic categories and spans a broad spectrum of motion intensities, providing a comprehensive benchmark for evaluating model performance.

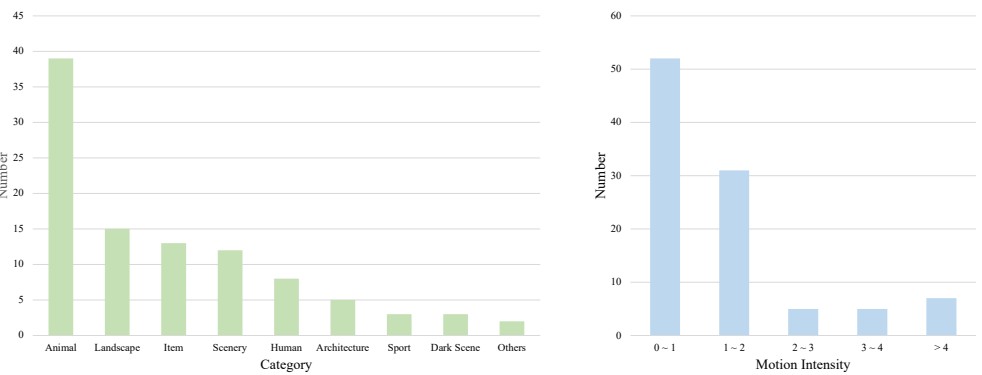

Figure 15: Statistical analysis of categories and motion in AIGC100

### A.7 PERFORMANCE ON REAL-WORLD LOW-QUALITY VIDEOS

The motivation of proposing SimpleGVR is to do computational decomposition in high-resolution T2V tasks. As such, SimpleGVR is lightweight, and its components are specifically tailored to our chosen T2V model. Evaluating its generalization to other datasets is therefore beyond the primary focus of our work.

We evaluate the performance of SimpleGVR on real-world low-quality videos from VideoLQ (Chan et al., 2022b), and the visual results are shown in Fig. 16. It can be observed that SimpleGVR is capable of removing certain degradations and recovering some details.

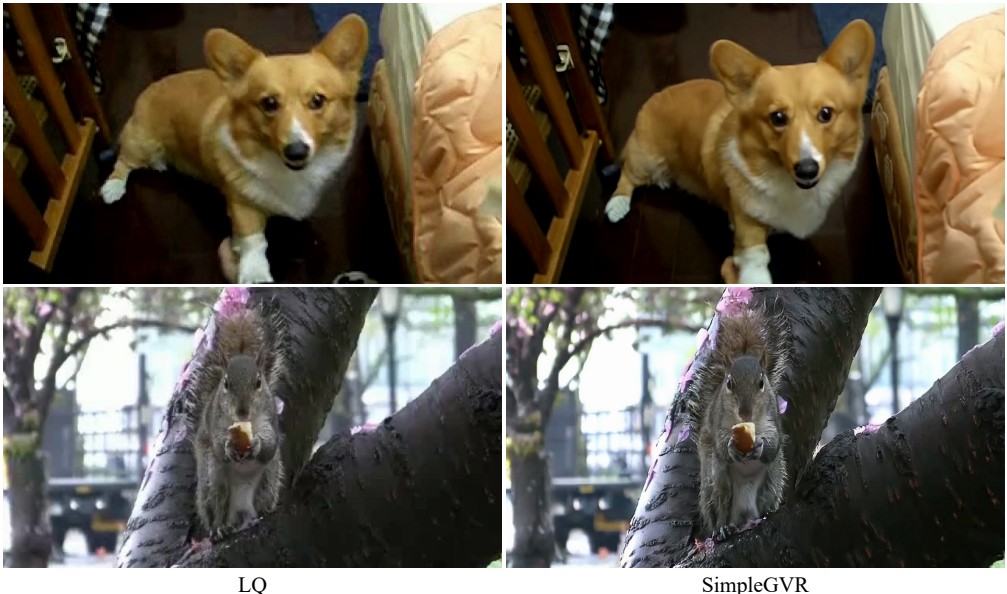

LQ                                    SimpleGVR

Figure 16: Qualitative results of SimpleGVR on the VideoLQ dataset.

### A.8 PERFORMANCE ON THE OUTPUT OF OTHER BASE T2V MODEL

Under the computational decoupling concept, SimpleGVR is proposed to play the role of cooperating with one specific base T2V model, instead of targeting for general video super-resolution task.

Table 7: Quantitative results of applying the pretrained SimpleGVR to low-resolution outputs generated by the Wan model.

| Method | MUSIQ | MANIQA | NIQE(↓) | CLIPIQA | DOVER | | | VBench Metrics | | | | | |
| | | | | | Technical | Aesthetic | Overall | Background Consistency | Subject Consistency | Aesthetic Quality | Imaging Quality | Motion Smoothness | Average Score |
|---|---|---|---|---|---|---|---|---|---|---|---|---|---|
| Wan-14B-480P | 64.76 | 0.365 | 4.456 | 0.559 | 13.94 | 96.90 | 55.69 | 95.35 | 95.44 | 62.33 | 68.35 | 98.93 | 84.08 |
| Wan-14B-720P | 68.55 | 0.441 | **4.320** | 0.612 | 17.87 | 97.18 | 60.78 | 95.04 | 94.95 | 62.09 | 69.11 | 98.57 | 83.95 |
| Wan-14B-480P + SimpleGVR | **70.05** | **0.463** | 4.328 | **0.641** | 18.61 | 97.33 | 62.51 | 95.01 | 95.35 | 62.06 | 70.86 | 98.87 | **84.43** |

Table 8: Quantitative results of applying the pretrained SimpleGVR to low-resolution outputs generated by the CogVideoX model.

| Method | MUSIQ | MANIQA | NIQE(↓) | CLIPIQA | DOVER | | | VBench Metrics | | | | | |
| | | | | | Technical | Aesthetic | Overall | Background Consistency | Subject Consistency | Aesthetic Quality | Imaging Quality | Motion Smoothness | Average Score |
|---|---|---|---|---|---|---|---|---|---|---|---|---|---|
| CogVideoX-5B-480×720 | 46.3907 | 0.2350 | 6.201 | 0.3202 | 9.82 | 96.00 | 46.37 | 95.33 | 94.50 | 59.25 | 62.15 | 97.91 | 81.83 |
| CogVideoX-V1.5-768×1360 | 53.4747 | 0.2575 | 4.893 | 0.3994 | 10.99 | 95.49 | 47.73 | 96.01 | 96.10 | 57.01 | 65.15 | 98.29 | 82.51 |
| CogVideoX-5B-480×720 + SimpleGVR | **66.0243** | **0.3413** | **4.064** | **0.5336** | 16.16 | 97.33 | 59.12 | 94.48 | 94.08 | 58.33 | 71.87 | 97.53 | **83.23** |

The rational lies that we require the tailor-made SimpleGVR to be as efficient as possible, so its model capacity is relatively lightweight, which is impractical to be expected to tackle general low-resolution video pattern. Anyway, the approach of SimpleGVR itself is general to any T2V base model, except that we need to adjust some hyper-parameters of flow-based degradation for training data preparation.

As an interesting extended study, we directly apply our trained SimpleGVR to the low-resolution inputs (i.e., 480p) generated by Wan-14B (Wan et al., 2025) and upscale them to 1080p high-resolution videos. The quantitative results are shown in Tab. 7. On most quantitative metrics, Wan-14B-480p + SimpleGVR achieves better performance, indicating that SimpleGVR can generalize to other base T2V models to some extent. As illustrated in Fig. 17, SimpleGVR is able to enhance details for the low-resolution inputs generated by Wan-14B.

We also conduct corresponding experiments on CogVideoX (Yang et al., 2024b). Specifically, we apply SimpleGVR directly to the 480×720 videos generated by CogVideoX-5B, producing 1080p outputs. The quantitative results are shown in Tab. 8. On most evaluation metrics, the combination of "CogVideoX-480×720 + SimpleGVR" achieves better performance. This observation is consistent with our findings on Wan-14B. As illustrated in Fig. 18, SimpleGVR is also capable of enhancing fine details for the low-resolution inputs generated by CogVideoX-5B.

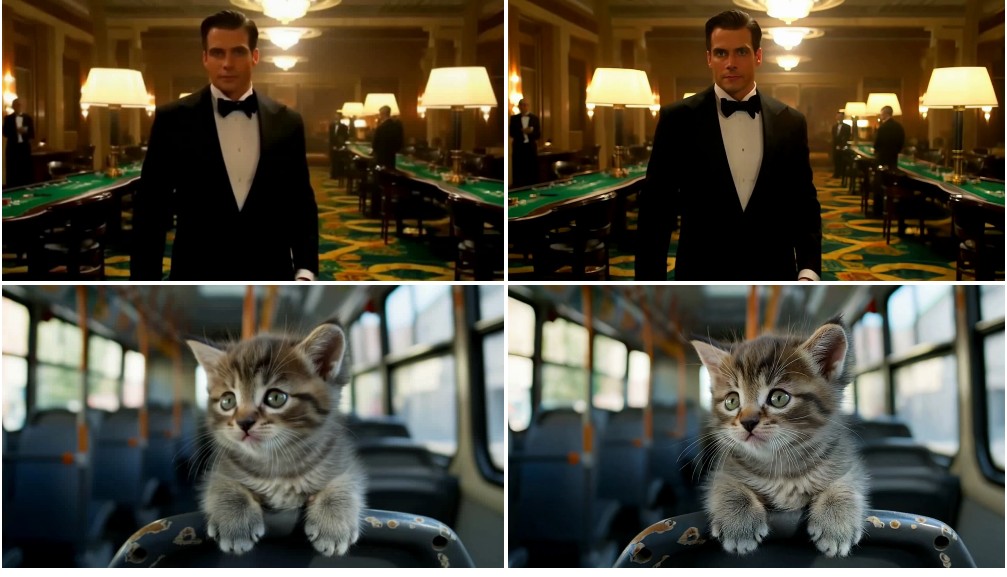

480p input from Wan-14B    1080p result with SimpleGVR

Figure 17: Qualitative results of SimpleGVR on the low-resolution output from Wan-14B.

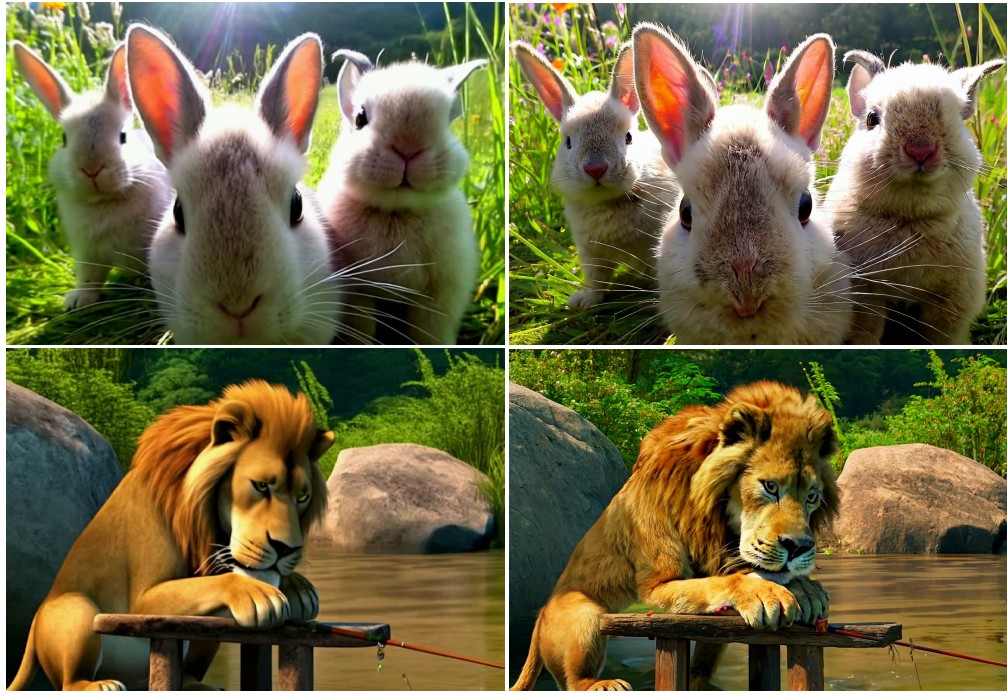

480×720 input from CogVideoX            1080p result with SimpleGVR

Figure 18: Qualitative results of SimpleGVR on the low-resolution output from CogVideoX.

### A.9 LIMITATION

As SimpleGVR is lightweight and primarily focusing on detail enhancement, it relies on the motion and overall structure of the low-resolution latent. Therefore, if the low-resolution latent is misaligned with the prompt or exhibits severe motion degradation, such issues will inevitably propagate to the final high-resolution video.

### A.10 DISCUSSIONS

**Why local blur and color blending artifacts appear?** It is common to see motion blur in real-world videos, which as training data, is challenging for T2V model to synthesis, especially when compressed by temporal VAE. So, the high-dynamic regions of generated video tend to suffer from abnormal blur and color blending.

**For high-resolution T2V generation target, why is fidelity to the low-resolution latent important?** The aim of our proposed two-stage high-resolution T2V framework is for computation decoupling. As the modeling of motion and structure is complex and requires large capacity, it is completed by the large base T2V model and inherited by SimpleGVR. Relaxing this fidelity constraint would challenge the lightweight SimpleGVR model to synthesize reasonable motion, structure, thereby obeying the purpose of decoupling. However, at minor structure/details level, SimpleGVR still has certain freedom to generate natural details. This fidelity&generation balance can be controlled by the noise added to the low-resolution latent.

## B THE USE OF LARGE LANGUAGE MODELS (LLMS)

In this work, we utilized large language models (LLMs) exclusively for the purpose of grammar checking and text polishing. Specifically, LLMs were employed to assist in enhancing the clarity, coherence, and readability of the text, by identifying and correcting grammatical errors, improving sentence structure, and refining language usage. These models were not involved in any aspect of the research ideation, data analysis, experimental design, or any other stages of the research process. The content, ideas, and conclusions presented in this work are solely the result of the authors' intellectual contributions.

