# OpenReview forum: "SimpleGVR: A Simple Baseline for Latent-Cascaded Generative Video Super-Resolution"
_ICLR.cc/2026/Conference — ICLR 2026 Poster_

### Official Review · Reviewer_w3HT · 2025-10-28

**Soundness:** 3
**Presentation:** 3
**Contribution:** 2
**Rating:** 6
**Confidence:** 5

**Summary:**

The paper proposes an algorithm, SimpleGVR, for Video Super-Resolution (VSR) tailored for AIGC videos. SimpleGVR enhances the resolution of a generated video by operating directly in the latent space. The main contributions of the paper includes: (a) A simple, lightweight yet effective model for VSR for videos generated by latent diffusion models, (b) New degradation strategies to simulate low-resolution synthetic videos, and (c) new sampling scheduler and noise augmentation for improving the low-res latent codes.

**Strengths:**

### Motivation
- Studying on efficient high-resolution video generation is critical for efficient video generation.

### Method
- The paper shows a simple yet efficient pipeline for AIGC video enhancement.
- The paper shares interesting insights into degradation for AIGC videos and a new framework for upsampling latent codes.

### Experimental results
- The authors show extensive experimental evaluations, especially their model designs.
- SimpleGVR outperforms many baselines over common benchmarks.
- SimpleGVR enables long-clip training/inference (77 frames) under limited GPUs.

### Writing/Presentation
The paper is well-written and easy-to-follow.

**Weaknesses:**

### Motivation
- Although reasonable and intuitive, the core idea (cascaded pipeline) is not new and exists in many video generation papers, such as Imagen-video from Google.

### Experimental results
- Temporal consistency is not well evaluated. Vbench metrics shown in the paper cannot capture temporal inconsistency.
- As many no-reference visual assessment metrics are not convincing, it is highly recommended a user study is conducted for more direct comparison.

**Questions:**

- Although only trained for AIGC videos, will SimpleGVR work for generic low-res videos by encoding real low-res videos into the latent space?

---

> ### Author Response · Authors · 2025-11-24
> **Official Comment by Authors**
>
> We are deeply grateful for the reviewer’s thoughtful insights and thorough evaluation of our manuscript. Please find what below our detailed point-to-point responses to all the comments of this reviewer. We hope our responses can address this reviewer's concerns.
>
> ---
>
> `[W1]: The cascade pipeline is not new and exists in many video generation papers.`
>
> The cascade pipeline is indeed a common paradigm for T2V generation. Our focus, however, is on how to achieve effective computational decoupling within this framework while improving the quality of the final high-resolution videos. To this end, we: 1) Propose a non-trivial latent upsampler that removes the extra decoding and re-encoding steps typically introduced in existing cascade pipeline methods. 2) Design two AIGC-centric degradation strategies to better mimic the output characteristics of the upstream T2V model, thereby enhancing performance. 3) Optimize the training configuration of SimpleGVR in three key aspects.
>
> Collectively, these contributions distinguish SimpleGVR from existing methods
>
> ---
> `[W2]: Temporal consistency is not well evaluated.`
>
> To better assess the temporal consistency of the super-resolved videos, we have conducted the user study in which participants were asked to select the video with the best temporal consistency, or equivalently, the lowest temporal flickering (lines 859-905). As shown in Figure 12(b) and Figure 13(b) of the Appendix, compared to other methods, more participants considered SimpleGVR to provide better temporal consistency.
>
> In Table 2 of the main paper, we additionally include the warp error metric to quantitatively assess temporal consistency. Please refer to it for details.
>
> ---
> `[W3]: A user study is recommend.`
>
> Thank you for the suggestion. We previously conducted user studies evaluating several methods, with results presented in Section A.4 of the Appendix. Since we have now included more recent baselines, we conducted a new round of user studies as well (lines 887–906). Among all compared approaches, SimpleGVR is the most preferred by participants.
>
> ---
>
> `[Q1]: Will SimpleGVR work for generic low-res videos?`
>
> As shown in Figure 16 of the Appendix, SimpleGVR can remove some degradations and recover certain details in real-world low-quality videos.

---

### Official Review · Reviewer_ExS4 · 2025-10-30

**Soundness:** 3
**Presentation:** 3
**Contribution:** 2
**Rating:** 4
**Confidence:** 4

**Summary:**

This paper introduces SimpleGVR, a lightweight video super-resolution method designed for cascaded text-to-video generation systems. The approach performs super-resolution directly in the latent space, avoiding redundant encoding and decoding operations in pixel space. A latent upsampler is used to integrate low-resolution latent features, and a multi-step diffusion denoising process generates high-resolution latent representations. The authors further propose two degradation modeling strategies tailored to the characteristics of generative video content, along with several training techniques to enhance detail quality and temporal consistency. Experiments show that SimpleGVR achieves higher efficiency and comparable visual quality to existing multi-stage cascaded methods on their constructed datasets.

**Strengths:**

1.The overall structure and motivation of the paper are reasonable. Performing video super-resolution in the latent space effectively avoids redundant VAE encoding and decoding, thereby reducing computational overhead. This idea is well justified within the cascaded text-to-video generation framework.

2.The experimental design includes ablation studies and hyperparameter analysis, which systematically demonstrate the effects of different components such as degradation modeling, latent injection strategies, and timestep sampling methods.

**Weaknesses:**

1.The work exhibits limited originality. The core idea of latent-space video super-resolution closely resembles prior studies such as FlashVideo, LaVie, and SeedVR. The proposed latent upsampler, channel concatenation, noise range setting, and timestep sampling strategy are largely incremental refinements or empirical combinations of existing approaches, lacking substantial algorithmic or theoretical innovation.

2.The experiments are conducted on the authors’ self-constructed AIGC100 and VBench110 datasets, which are not publicly available, making it difficult to assess fairness and reproducibility. The comparison settings are not fully aligned, as details about the base T2V model, training data, and inference steps are not clearly specified.

3.The paper does not compare with stronger diffusion-based VSR methods such as DiffVSR or VideoGigaGAN, nor does it evaluate whether the proposed model generalizes to different upstream T2V generators such as CogVideoX.

**Questions:**

1.The paper states that injecting noise into the low-resolution branch helps the model correct structural errors and recover fine details, but the underlying mechanism is not well explained. It is recommended that the authors clarify whether the noise distribution is consistent between training and inference, and whether any mismatch could affect model stability. The paper also claims that the model can “correct structural errors,” but the limits of this ability are unclear. Current experiments only show mild distortions; more extreme cases would better demonstrate robustness.

2.In the degradation modeling section, the flow-based degradation uses optical flow and blur to mimic artifacts observed in generated videos, yet it remains unclear whether these synthetic artifacts truly resemble those of real T2V outputs. The authors are encouraged to provide quantitative comparisons, such as spectral statistics or color-blending analysis. In the model-guided degradation strategy, where the upstream generator is involved in synthesizing training data, there is a risk that SimpleGVR might merely learn the biases of the teacher model rather than general restoration capability. A cross-model experiment, such as training on data from model A and testing on model B, would help evaluate generalization.

3.The AIGC100 test set is manually curated by the authors, but the selection criteria and data distribution are not disclosed. It is recommended to report the category composition, motion intensity distribution, or include evaluations on public benchmarks to enhance credibility.

4.The evaluation currently relies solely on no-reference metrics such as MUSIQ, DOVER, and VBench, without subjective assessment or explicit temporal consistency measures. Incorporating human perceptual scores (e.g., MOS) or motion-consistency metrics would provide a more comprehensive evaluation of video quality.

---

> ### Author Response · Authors · 2025-11-24
> **Official Comment by Authors (Part 1/5)**
>
> We sincerely appreciate this reviewer's insightful feedback and careful review of our manuscript. Please find what below our detailed point-to-point responses to all the comments of this reviewer. We hope our responses can address this reviewer's concerns.
>
> ---
>
> `[W1]: The work exhibits limited originality, since the core idea of latent-space video super-resolution closely resembles prior studies such as FlashVideo, LaVie, and SeedVR.`
>
> FlashVideo, Lavie, and SeedVR do not perform upscaling directly on low-resolution latents. Instead, they require decoding the low-resolution latent into RGB video, applying upscaling in pixel space, and then re-encoding the video back into latent space—introducing substantial additional computational cost. To avoid these redundant decoding and re-encoding steps, we design a latent upsampler that operates in latent space with negligible computation burden. Its technical novelty and importance is clarified in the common response section.
>
>
> Furthermore, within the cascade pipeline, we perform more fine-grained data construction and propose two degradation strategies to mimic the output characteristics of the upstream T2V model.
>
> Together, these design choices make SimpleGVR stand out among existing methods and highlight its unique contribution to the community.
>
> ---
>
> `[W2] Regarding the reproducibility and fairness of experiment settings.`
>
> Thank you for this insightful comment. Like many works leveraging large foundation models, a strictly fair comparison with all existing methods is computationally prohibitive, requiring extensive re-training across various models and datasets.
>
> Nonetheless, we believe our experimental results are informative and meaningful. Our SimpleGVR, despite being significantly more lightweight (1B parameters) than existing VSR methods, achieves notable superiority. This performance gain primarily stems from our tailor-made data synthesis pipeline, which is meticulously aligned with our T2V base model. This is exactly what this paper target for, i.e. a ligntweight VSR model tailor-made to cooperate with one specific base model, instead of targeting for general VSR task. Besides, the abalation study confirms the effectiveness of invidual designs in fully aligned settings.

---

> ### Author Response · Authors · 2025-11-24
> **Official Comment by Authors (Part 2/5)**
>
> `[W3]: Compare with stronger diffusion-based VSR methods.`
>
> Thank you for the suggestion. DiffVSR[1] had not released its code before our ICLR submission (as confirmed on its GitHub), and VideoGigaGAN[2] remains closed-source to date. Therefore, these methods could not be included during the submission period. Now the DiffVSR is available and is included in our baselines, in addition, we additionally incorporate several recent and representative approaches—MGLD [3], DLoRAL [4], DOVE [5], and SeedVR2 [6]—to enable a more comprehensive and convincing performance comparison. The results are presented below and have been incorporated into Table 2 of the main paper.
>
> |     Methods     |   MUSIQ   |   CLIPIQA  |   MANIQA   |   NIQE(↓)  | $E^{*}_{warp}\times10^{3}$(↓) | DOVER-Technical | DOVER-Aesthetic | DOVER-Overall | Background Consistency | Subject  Consistency | Aesthetic Quality | Imaging Quality | Motion Smoothness | VBench-Average |
> |:---------------:|:---------:|:----------:|:----------:|:----------:|:---------------------:|:---------------:|:---------------:|:-------------:|:----------------------:|:--------------------:|:-----------------:|:---------------:|:-----------------:|:--------------:|
> |   RealBasicVSR  |   57.55   |   0.5970   |   0.4591   |   4.6062   |         4.485         |      12.27      |      98.66      |     61.84     |          93.73         |         93.98        |       61.63       |      72.76      |       98.70       |      84.16     |
> |    VEnhancer    |   40.03   |   0.5034   |   0.3429   |   5.3319   |         2.796         |      15.38      |      98.32      |     62.54     |          94.59         |         94.44        |       59.98       |      64.22      |       99.16       |      82.48     |
> | Upscale-A-Video |   36.35   |   0.4744   |   0.3033   |   5.7165   |         4.314         |      12.43      |      98.29      |     59.04     |          95.96         |         94.41        |       61.26       |      63.85      |       98.99       |      82.89     |
> |       STAR      |   46.73   |   0.5469   |   0.3743   |   4.9787   |       **2.409**       |      18.17      |      98.66      |     67.76     |          96.17         |         94.43        |       62.24       |      67.24      |       99.01       |      83.82     |
> |    Flashvideo   |   53.71   |   0.5818   |   0.4262   |   4.8130   |         4.314         |      17.51      |      98.61      |     67.35     |          96.14         |         95.14        |       61.94       |      68.04      |       98.72       |      84.00     |
> |   SeedVR (7B)   |   56.77   |   0.6176   |   0.4328   |   4.3025   |         3.800         |      18.05      |      97.40      |     61.87     |          94.80         |         93.80        |       63.82       |      69.49      |       98.51       |      84.08     |
> |   SeedVR2 (7B)  |   53.51   |   0.6179   |   0.4242   |   4.3552   |         3.814         |      17.71      |      97.51      |     61.88     |          94.84         |         93.80        |       63.63       |      69.42      |       98.55       |      84.05     |
> |       DOVE      |   60.34   |   0.5982   |   0.4332   |   4.7323   |         3.180         |      16.81      |      97.63      |     61.54     |          96.89         |         94.02        |       62.98       |      69.17      |       98.87       |      84.39     |
> |       MGLD      |   52.19   |   0.6142   |   0.4260   |   4.1880   |         3.877         |      12.62      |      97.59      |     56.97     |          96.21         |         94.61        |       61.57       |      70.96      |       98.67       |      84.40     |
> |      DLoRAL     |   58.57   |   0.5975   |   0.4302   |   4.5683   |         3.704         |      14.23      |      97.61      |     58.36     |          95.84         |         94.07        |       64.21       |      69.20      |       98.63       |      84.39     |
> |     DiffVSR     |   55.32   |   0.5618   |   0.4074   |   4.7305   |         3.415         |      14.77      |      97.76      |     54.97     |          95.42         |         96.82        |       62.14       |      68.09      |       98.80       |      84.25     |
> |       Ours      | **62.35** | **0.6768** | **0.4956** | **4.1665** |         2.592         |    **20.44**    |    **98.88**    |   **71.34**   |          95.35         |         94.32        |       62.84       |      71.91      |       98.74       |    **84.63**   |
>
> ***

---

> ### Author Response · Authors · 2025-11-24
> **Official Comment by Authors (Part 3/5)**
>
> `[W3]: Evaluate the generalization ability to different upstream T2V generators.`
>
> As you suggested, to evaluate the generalization ability of SimpleGVR, we directly apply SimpleGVR to the low-res inputs (i.e., 480p) generated by Wan-14B[7]. The quantitative results are shown below, and qualitative results can be found in Figure 17 in the Appendix. On most quantitative metrics, Wan-14B-480p + SimpleGVR achieves better performance, indicating that SimpleGVR can generalize to other base T2V models to some extent.
>
> |          Models          |   MUSIQ   |   MANIQA  |    NIQE(↓)   |  CLIP-IQA | DOVER-Techinical | DOVER-Aesthetic | DOVER-Overall | Background Consistency | Subject  Consistency | Aesthetic Quality | Imaging Quality | Motion Smoothness | VBench-Average |
> |:------------------------:|:---------:|:---------:|:---------:|:---------:|:----------------:|:---------------:|:-------------:|:----------------------:|:--------------------:|:-----------------:|:---------------:|:-----------------:|:--------------:|
> |       Wan-14B-480P       |   64.76   |   0.365   |   4.456   |   0.559   |       13.94      |      96.90      |     55.69     |          95.35         |         95.44        |       62.33       |      68.35      |       98.93       |      84.08     |
> |       Wan-14B-720P       |   68.55   |   0.441   | **4.320** |   0.612   |       17.87      |      97.18      |     60.78     |          95.04         |         94.95        |       62.09       |      69.11      |       98.57       |      83.95     |
> | **Wan-14B-480P + SimpleGVR** | **70.05** | **0.463** |   4.328   | **0.641** |     **18.61**    |    **97.33**    |   **62.51**   |          95.01         |         95.35        |       62.06       |      70.86      |       98.87       |    **84.43**   |
>
> We also conduct corresponding experiments on CogVideoX[8]. Specifically, we apply SimpleGVR directly to the 480×720 videos generated by CogVideoX-5B, producing 1080p outputs. The quantitative results are shown below, and qualitative results can be found in Figure 18 in the Appendix. On most quantitative metrics, ''CogVideoX-480$\times$720 + SimpleGVR'' achieves better performance, indicating that SimpleGVR can generalize to CogVideoX to some extent.
>
> |           Method          |    MUSIQ    |   MANIQA   |  NIQE(↓)  |   CLIPIQA  | DOVER-Techinical | DOVER-Aesthetic | DOVER-Overall | Background Consistency | Subject  Consistency | Aesthetic Quality | Imaging Quality | Motion Smoothness | VBench-Average  |
> |:-------------------------:|:-----------:|:----------:|:---------:|:----------:|:----------------:|:---------------:|:-------------:|:----------------------:|:--------------------:|:-----------------:|:---------------:|:-----------------:|:---------------:|
> |       CogVideoX-5B-480$\times$720        |   46.3907   |   0.2350   |   6.201   |   0.3202   |       9.82       |      96.00      |     46.37     |          95.33         |         94.50        |       59.25       |      62.15      |       97.91       |      81.83      |
> |       CogVideoX-V1.5-768$\times$1360       |   53.4747   |   0.2575   |   4.893   |   0.3994   |       10.99      |      95.49      |     47.73     |          96.01         |         96.10        |       57.01       |      65.15      |       98.29       |      82.51      |
> | CogVideoX-5B-480$\times$720 + SimpleGVR | **66.0243** | **0.3413** | **4.064** | **0.5336** |     **16.16**    |    **97.33**    |     59.12     |          94.48         |         94.08        |       58.33       |      71.87      |       97.53       |    **83.23**    |
>
> ***
> `[Q1]: Whether the noise distribution in low-resolution branch is consistent between training and inference? Limitations of noise augmeentation's ability to correct structural errors.`
>
> The purpose of adding noise augmentation during joint training is to weaken the influence of the LR condition and thereby enhance the generative capability of the VSR model. For this reason, we adopt a fixed Gaussian distribution, which is kept **consistent during both training and testing**.
>
> The capability of SimpleGVR to correct structural errors primarily relies on the detail synthesis capacity of the VSR module. Incorporating noise augmentation in the low-resolution branch serves to relax the strong constraints imposed by the LR input, enhancing the generative capability of the VSR module. Nevertheless, as a lightweight model, SimpleGVR’s corrective capacity is inherently limited: it primarily addresses fine-grained details and does not correct semantic inconsistencies or large-scale structural errors. Addressing semantic inconsistencies or large-scale structural errors falls within the scope of the base T2V model.

---

> ### Author Response · Authors · 2025-11-24
> **Official Comment by Authors (Part 4/5)**
>
> `[Q2]: The authors are encouraged to provide quantitative comparisons, such as spectral statistics or color-blending analysis.`
>
> Thank you for the suggestion. However, the flow-based degradation is adopted to synthesize low-resolution videos from real-world high-quality videos, whereas our goal is to mimic the artifacts present in prompt-generated AIGC videos. These two types of videos (synthesized low-resolution videos and AIGC videos) exhibit significant content differences because of generative randomness, making it difficult to perform a direct similarity analysis of artifacts. Therefore, we can only rely on experimental results to indirectly assess whether the proposed degradation strategy effectively mimics the artifacts observed in generated videos. As shown in Table 4 in the main paper, the flow-based degradation demonstrates its effectiveness.
>
> `[Q2]: A cross-model experiment would help evaluate generalization.`
>
> The motivation of proposing SimpleGVR is to do computational decomposition in high-resolution T2V tasks. As such, SimpleGVR is lightweight, and its components are specifically tailored to our chosen T2V model. Evaluating its generalization to other datasets is therefore beyond the primary focus of our work. Following your suggestion, we directly applied SimpleGVR to 480p outputs generated by Wan-14B, CogVideoX, as well as to the VideoLQ restoration dataset. The results indicate that SimpleGVR exhibits a certain degree of generalization capability. For details on the Wan-14B and the CogVideoX results, please refer to Section A.9 of the Appendix; for the VideoLQ results, see Section A.10.
>
> `[Q3]: It is recommended to report the category composition, motion intensity distribution of AIGC100.`
>
> AIGC100 is our internal evaluation dataset, consisting of low-resolution videos generated by the base T2V model. In Figure 15 of the Appendix, we present its category composition and motion intensity distribution.

---

> ### Author Response · Authors · 2025-11-24
> **Official Comment by Authors (Part 5/5)**
>
> `[Q4]: Subjective assessment or explicit temporal consistency measurement should be adopted.`
>
> A: Thank you for the suggestion. We previously conducted user studies evaluating several methods, with results presented in Section A.4 of the Appendix. Since we have now included additional baselines, we conducted a new round of user studies as well (lines 887–906). Among all compared approaches, SimpleGVR is the most preferred by participants.
>
> For motion consistency, following DOVE and DiffVSR, we adopt the $E^{*}_{warp}$ metric. As shown in the table below, SimpleGVR also achieves strong performance on this metric. We have incorporated the results into Table 2 of the main paper.
>
>
> |     Methods     |   MUSIQ   |   CLIPIQA  |   MANIQA   |   NIQE(↓)  | $E^{*}_{warp}\times10^{3}$(↓) | DOVER-Technical | DOVER-Aesthetic | DOVER-Overall | Background Consistency | Subject  Consistency | Aesthetic Quality | Imaging Quality | Motion Smoothness | VBench-Average |
> |:---------------:|:---------:|:----------:|:----------:|:----------:|:---------------------:|:---------------:|:---------------:|:-------------:|:----------------------:|:--------------------:|:-----------------:|:---------------:|:-----------------:|:--------------:|
> |   RealBasicVSR  |   57.55   |   0.5970   |   0.4591   |   4.6062   |         4.485         |      12.27      |      98.66      |     61.84     |          93.73         |         93.98        |       61.63       |      72.76      |       98.70       |      84.16     |
> |    VEnhancer    |   40.03   |   0.5034   |   0.3429   |   5.3319   |         2.796         |      15.38      |      98.32      |     62.54     |          94.59         |         94.44        |       59.98       |      64.22      |       99.16       |      82.48     |
> | Upscale-A-Video |   36.35   |   0.4744   |   0.3033   |   5.7165   |         4.314         |      12.43      |      98.29      |     59.04     |          95.96         |         94.41        |       61.26       |      63.85      |       98.99       |      82.89     |
> |       STAR      |   46.73   |   0.5469   |   0.3743   |   4.9787   |       **2.409**       |      18.17      |      98.66      |     67.76     |          96.17         |         94.43        |       62.24       |      67.24      |       99.01       |      83.82     |
> |    Flashvideo   |   53.71   |   0.5818   |   0.4262   |   4.8130   |         4.314         |      17.51      |      98.61      |     67.35     |          96.14         |         95.14        |       61.94       |      68.04      |       98.72       |      84.00     |
> |   SeedVR (7B)   |   56.77   |   0.6176   |   0.4328   |   4.3025   |         3.800         |      18.05      |      97.40      |     61.87     |          94.80         |         93.80        |       63.82       |      69.49      |       98.51       |      84.08     |
> |   SeedVR2 (7B)  |   53.51   |   0.6179   |   0.4242   |   4.3552   |         3.814         |      17.71      |      97.51      |     61.88     |          94.84         |         93.80        |       63.63       |      69.42      |       98.55       |      84.05     |
> |       DOVE      |   60.34   |   0.5982   |   0.4332   |   4.7323   |         3.180         |      16.81      |      97.63      |     61.54     |          96.89         |         94.02        |       62.98       |      69.17      |       98.87       |      84.39     |
> |       MGLD      |   52.19   |   0.6142   |   0.4260   |   4.1880   |         3.877         |      12.62      |      97.59      |     56.97     |          96.21         |         94.61        |       61.57       |      70.96      |       98.67       |      84.40     |
> |      DLoRAL     |   58.57   |   0.5975   |   0.4302   |   4.5683   |         3.704         |      14.23      |      97.61      |     58.36     |          95.84         |         94.07        |       64.21       |      69.20      |       98.63       |      84.39     |
> |     DiffVSR     |   55.32   |   0.5618   |   0.4074   |   4.7305   |         3.415         |      14.77      |      97.76      |     54.97     |          95.42         |         96.82        |       62.14       |      68.09      |       98.80       |      84.25     |
> |       Ours      | **62.35** | **0.6768** | **0.4956** | **4.1665** |         2.592         |    **20.44**    |    **98.88**    |   **71.34**   |          95.35         |         94.32        |       62.84       |      71.91      |       98.74       |    **84.63**   |
>
> ***

---

> > ### Comment · Reviewer_ExS4 · 2025-11-27
> > **Reply**
> >
> > Thank you for the detailed clarifications and the additional experiments. I believe your responses adequately address the main concerns raised in my earlier review. After further consideration, I am willing to raise my score to 6 in recognition of the improvements and supplementary results provided.

---

> > > ### Author Response · Authors · 2025-11-27
> > > **Official Comment by Authors**
> > >
> > > Dear Reviewer  ExS4,
> > >
> > > We are truly encouraged and grateful for your positive feedback and your decision to raise the rating to 6.
> > >
> > > We are delighted to learn that our detailed response and the new experimental results have successfully resolved your main concerns. Your constructive feedback throughout the review process has been invaluable in helping us identify areas for improvement and strengthen the validation of our method.
> > >
> > > Thank you once again for your time, support, and contribution to improving our work.
> > >
> > > Best regards,
> > >
> > > The Authors of Submission 5131

---

> ### Author Response · Authors · 2025-11-27
> **Follow-up on Rebuttal Response for Paper 5131**
>
> Dear Reviewer ExS4,
>
> We hope this message finds you well.
>
> As the discussion period is coming to a close, we would like to kindly ensure that our response and the **extensive new experiments** conducted based on your constructive suggestions have reached you.
>
> In our rebuttal, we have made significant updates to address your concerns:
>
> * **Clarification on Originality (W1):** We clarified the distinction between our latent upsampler and methods like FlashVideo/LaVie. Our approach operates directly in the latent space to avoid the heavy computational cost of decoding-encoding. Furthermore, within the cascade pipeline, we perform more fine-grained data construction and propose two degradation strategies to mimic the output characteristics of the upstream T2V model.
> * **Generalization to Other T2V Models (W3 & Q2):** To demonstrate the generalization ability of our approach, we directly applied our trained SimpleGVR to **Wan-14B** and **CogVideoX**. The results (detailed in the Appendix, Table 7, Figure 17, Table 8, Figure 18) confirm that our trained SimpleGVR can generalize to other T2V models.
> * **Comparison with Stronger Baselines (W3):** Following your suggestion, we have incorporated **5+ additional state-of-the-art methods**, including **DiffVSR, MGLD, DLoRAL, DOVE, and SeedVR2**. As shown in the updated Table 2, our method consistently outperforms these strong baselines across most metrics.
> * **Additional Evaluations (Q4):** We added the temporal consistency metric ($E_{warp}$) and conducted a new **User Study**, which further validates the superiority of our method.
>
> Given the limited time remaining in the discussion period, we would be extremely grateful if you could verify whether our extensive new experiments and clarifications help resolve your initial concerns. **If you find that our response and these new results have successfully strengthened the paper and addressed your questions, we would highly appreciate it if you could consider raising your score to reflect these significant improvements.**
>
> Thank you again for your time and valuable input to improve our work.
>
> We remain available to answer any further questions.
>
> Best regards,
>
> The Authors of Submission 5131

---

### Official Review · Reviewer_D3ZN · 2025-10-31

**Soundness:** 2
**Presentation:** 2
**Contribution:** 3
**Rating:** 6
**Confidence:** 4

**Summary:**

The paper aims to address the heavy computational cost and suboptimal visual quality of AIGC videos. It proposes a method named SimpleGVR, and designs explicit degradation pipelines to match the distribution of AIGC videos. Compared to existing VSR methods, it is interesting to consider the VSR problem within the paradigm of video generation.

**Strengths:**

(1)	The observations of “video generation + vsr” pipeline are interesting. There are two key observations, architectural redundancy and degradation misalignment. Both of them are key to high-quality video generation.

(2)	The synthesis pipeline of LR videos is quite reasonable, since it considers the distribution of T2V model.

**Weaknesses:**

(1)	The motivation for the two synthetic degradations is unclear. Figure 4, used to justify common degradations in AIGC videos, is unconvincing. For example, the color blending example, which is very similar to natural lighting or shadow variation and seems to have little impact on video quality. Providing clearer, more representative cases where these degradations noticeably degrade perceptual quality would strengthen the rationale for the subsequent degradation design.

(2)	The evaluation is limited in both dataset and metrics. The paper adopts AIGC100 as the testing dataset, which aligns with the goal of enhancing AIGC videos but lacks diversity to verify the method’s generalization. Since AIGC100 contains AI-generated videos whose degradations differ from those in most compared models’ training data, it is unclear whether the improvement comes from the model itself or from the matched degradation distribution. Additional results on general datasets (e.g., VideoLQ) are suggested. As for evaluation metrics, besides MUSIQ and DOVER, it would be more convincing to include CLIPIQA, MANIQA, and Warping Error for a more comprehensive quality assessment.

(3)	The comparison methods are also limited. It would strengthen the experimental evaluation to include more recent or representative approaches, such as MGLD, DLoRAL, Dove, and SeedVR2, to provide a more comprehensive and convincing performance comparison.

**Questions:**

(1)	A large portion of the paper, including Figure 2, is devoted to describing the combination of LR and GT features. However, the conclusion drawn from this design seems rather intuitive, since the latent upsampler introduces additional 3D ResBlocks compared to latent interpolation. This part feels redundant and could be streamlined for clarity.

(2)	Figure 8 is hard to understand. The meaning of symbol “l” is not explained and annotations of “2k/2k+1/2k+2” are ambiguous. And also, what does “shift window” mean exactly?

---

> ### Author Response · Authors · 2025-11-24
> **Official Comment by Authors (Part 1/2)**
>
> We are deeply grateful for the reviewer’s thoughtful insights and thorough evaluation of our manuscript. Please find what below our detailed point-to-point responses to all the comments of this reviewer. We hope our responses can address this reviewer's concerns.
>
> ---
>
> `[W1] Providing clearer, more representative cases where these degradations noticeably degrade perceptual quality.`
>
> Thank you for the suggestion. We have updated Figure 4 in the main paper.
>
> ---
>
> `[W2] The evaluation dataset is limited, i.e. mainly on AIGC100 generated by internal base model.`
>
> We cannot agree with this perspective. Our SimpleGVR operates within a specific problem context: computational decoupling for high-resolution T2V generation. It is intentionally lightweight and uniquely trained to cooperate with a particular T2V base model, not to function as a general VSR solution for arbitrary low-resolution videos. Such a broad scope is both impractical for a lightweight model and irrelevant to our specific use case.
>
> Crucially, our SimpleGVR (1B) outperforms existing general VSR models on the AIGC100 dataset despite its significantly smaller size. This superior performance is a direct result of our tailored data preparation pipeline, which precisely matches the degradation characteristics of the base T2V model. This specialized design validates the effectiveness of our approach; it should be viewed as a demonstration of specialized excellence, not an unfair comparison.
>
> ---
>
> `[W2] The evaluation metrics is limited.`
>
> Following your suggestion, we additionally report CLIPIQA, MANIQA, and warp error to provide a more comprehensive evaluation across perceptual and temporal dimensions. SimpleGVR achieves the best performance on the majority of these metrics.

---

> ### Author Response · Authors · 2025-11-24
> **Official Comment by Authors (Part 2/2)**
>
> `[W3] Comparison with more recent or representative approaches (e.g., MGLD, DLoRAL, Dove, and SeedVR2).`
>
> Thank you for the suggestion. We have included the results of the additional methods you mentioned. As shown in the table below, SimpleGVR still achieves the best overall performance across the evaluated metrics. The corresponding results have also been updated in Table 2 of the main paper.
>
> |     Methods     |   MUSIQ   |   CLIPIQA  |   MANIQA   |   NIQE(↓)  | $E^{*}_{warp} \times 10^{3}$(↓) | DOVER-Technical | DOVER-Aesthetic | DOVER-Overall | Background Consistency | Subject  Consistency | Aesthetic Quality | Imaging Quality | Motion Smoothness | VBench-Average |
> |:---------------:|:---------:|:----------:|:----------:|:----------:|:---------------------:|:---------------:|:---------------:|:-------------:|:----------------------:|:--------------------:|:-----------------:|:---------------:|:-----------------:|:--------------:|
> |   RealBasicVSR  |   57.55   |   0.5970   |   0.4591   |   4.6062   |         4.485         |      12.27      |      98.66      |     61.84     |          93.73         |         93.98        |       61.63       |      72.76      |       98.70       |      84.16     |
> |    VEnhancer    |   40.03   |   0.5034   |   0.3429   |   5.3319   |         2.796         |      15.38      |      98.32      |     62.54     |          94.59         |         94.44        |       59.98       |      64.22      |       99.16       |      82.48     |
> | Upscale-A-Video |   36.35   |   0.4744   |   0.3033   |   5.7165   |         4.314         |      12.43      |      98.29      |     59.04     |          95.96         |         94.41        |       61.26       |      63.85      |       98.99       |      82.89     |
> |       STAR      |   46.73   |   0.5469   |   0.3743   |   4.9787   |       **2.409**       |      18.17      |      98.66      |     67.76     |          96.17         |         94.43        |       62.24       |      67.24      |       99.01       |      83.82     |
> |    Flashvideo   |   53.71   |   0.5818   |   0.4262   |   4.8130   |         4.314         |      17.51      |      98.61      |     67.35     |          96.14         |         95.14        |       61.94       |      68.04      |       98.72       |      84.00     |
> |   SeedVR (7B)   |   56.77   |   0.6176   |   0.4328   |   4.3025   |         3.800         |      18.05      |      97.40      |     61.87     |          94.80         |         93.80        |       63.82       |      69.49      |       98.51       |      84.08     |
> |   SeedVR2 (7B)  |   53.51   |   0.6179   |   0.4242   |   4.3552   |         3.814         |      17.71      |      97.51      |     61.88     |          94.84         |         93.80        |       63.63       |      69.42      |       98.55       |      84.05     |
> |       DOVE      |   60.34   |   0.5982   |   0.4332   |   4.7323   |         3.180         |      16.81      |      97.63      |     61.54     |          96.89         |         94.02        |       62.98       |      69.17      |       98.87       |      84.39     |
> |       MGLD      |   52.19   |   0.6142   |   0.4260   |   4.1880   |         3.877         |      12.62      |      97.59      |     56.97     |          96.21         |         94.61        |       61.57       |      70.96      |       98.67       |      84.40     |
> |      DLoRAL     |   58.57   |   0.5975   |   0.4302   |   4.5683   |         3.704         |      14.23      |      97.61      |     58.36     |          95.84         |         94.07        |       64.21       |      69.20      |       98.63       |      84.39     |
> |     DiffVSR     |   55.32   |   0.5618   |   0.4074   |   4.7305   |         3.415         |      14.77      |      97.76      |     54.97     |          95.42         |         96.82        |       62.14       |      68.09      |       98.80       |      84.25     |
> |       Ours      | **62.35** | **0.6768** | **0.4956** | **4.1665** |         2.592         |    **20.44**    |    **98.88**    |   **71.34**   |          95.35         |         94.32        |       62.84       |      71.91      |       98.74       |    **84.63**   |
>
> ***
>
> `[Q1]: The conclusion drawn from the latent upsampler seems rather intuitive.`
>
> The technical novelty and importance of latent upsampler is clarified in the common response section.
>
> ---
>
> `[Q2]: Figure 8 is hard to understand.`
>
> We apologize for the confusion. The symbol **“l”** denotes the set of all transformer blocks in DiT, and **“2k / 2k+1 / 2k+2”** refer to their corresponding indices, where $l_{2k}$ represents the *2k-th* transformer block. The term **“shift window”** denotes a Swin-style temporal shift, in which the attention window is moved by half its size along the temporal axis to enable cross-window temporal interactions. We have included these clarifications in the main paper (lines 350-360).

---

### Official Review · Reviewer_tr3b · 2025-11-01

**Soundness:** 3
**Presentation:** 3
**Contribution:** 3
**Rating:** 6
**Confidence:** 4

**Summary:**

This paper proposed a diffusion-based latent video super resolution model for T2V model outputs. It first inject the low-res latent into VSR model by a proposed latent upsampler and channel concatenation operations. To mitigate the gap between synthetic low-res latent and real T2V model output latents, the author proposed two degradation schemes, i.e., flow-based and model-guided degradation. Finally it also introduce multiple training receipts for improved performance, controllable generative details and long video super resolution.

**Strengths:**

- VSR directly performs on latent space, instead of introducing decoding and re-encoding from lossy and compute heavy VAE.
- There are comprehensive low-res latent injection methods and discussion, which gives a better understanding of the best low-res to high-res latent mapping.
- It proposed flow-based and model-guided degradation schemes to simulate the degradation characteristics of the T2V base model outputs.
- The detail-aware timestep sampler, optimal noise augmentation range, and interleaving temporal unit mechanism enhance the model generalization ability and robustness.

**Weaknesses:**

- The overall model design is specific to the base T2V model used. Specifically, the design of flow-based degradation is used to mimic the artifacts from the specific base model. It would be helpful to better understand the generalization ability of the proposed model on other baseline T2V models.
- The flow-based degradation though novel, may introduce multiple stages like optical flow estimation, color blending, distance-based weighting, etc. This can bring complexity of the pipeline and is likely prone to error.

**Questions:**

- For the proposed low-res latent injection method, it first increase the channel and spatial/time resolution and reduce them in the end. What is the rationale behind it? And is there any ablation about any simpler designs like directly increasing the channel and spatial/time resolution to match the high-res latents?
- How is the proposed detail-aware sampler implemented in details? Is it the shifted/warped timestep toward high noise level sampler?
- How to resolve the memory consumption when dealing with mulitple video chunk for each transformer layers? Are all intermediate results cached while computing so that they can be shifted high window?

---

> ### Author Response · Authors · 2025-11-24
> **Official Comment by Authors (Part 1/2)**
>
> We sincerely appreciate this reviewer's insightful feedback and careful review of our manuscript. Please check our detailed point-to-point responses to all the comments of this reviewer. We hope our responses can address this reviewer's concerns.
>
> ---
>
> `[W1] The generalization ability of SimpleGVR on other baseline T2V models.`
>
> Under the computational decoupling concept, SimpleGVR is proposed to play the role of cooperating with one specific base T2V model, instead of targeting for general video super-resolution task. The rational lies that we require the tailor-made SimpleGVR to be as efficient as possible, so its model capacity is relatively lightweight, which is impractical to be expected to tackle general low-resolution video pattern. Anyway, the approach of SimpleGVR itself is general to any T2V base model, except that we need to adjust some hyper-parameters of flow-based degradation for training data preparation.
>
> As an interesting extended study, we directly apply our trained SimpleGVR to the low-resolution inputs (i.e., 480p) generated by Wan-14B and upscale them to 1080p high-resolution videos. The quantitative results are shown below, and qualitative results can be found in Figure 17 of the Appendix. On most quantitative metrics, Wan-14B-480p + SimpleGVR achieves better performance, indicating that SimpleGVR can generalize to Wan to some extent.
>
> |          Models          |   MUSIQ   |   MANIQA  |    NIQE(↓)   |  CLIP-IQA | DOVER-Techinical | DOVER-Aesthetic | DOVER-Overall | Background Consistency | Subject  Consistency | Aesthetic Quality | Imaging Quality | Motion Smoothness | VBench-Average |
> |:------------------------:|:---------:|:---------:|:---------:|:---------:|:----------------:|:---------------:|:-------------:|:----------------------:|:--------------------:|:-----------------:|:---------------:|:-----------------:|:--------------:|
> |       Wan-14B-480P       |   64.76   |   0.365   |   4.456   |   0.559   |       13.94      |      96.90      |     55.69     |          95.35         |         95.44        |       62.33       |      68.35      |       98.93       |      84.08     |
> |       Wan-14B-720P       |   68.55   |   0.441   | **4.320** |   0.612   |       17.87      |      97.18      |     60.78     |          95.04         |         94.95        |       62.09       |      69.11      |       98.57       |      83.95     |
> | **Wan-14B-480P + SimpleGVR** | **70.05** | **0.463** |   4.328   | **0.641** |     **18.61**    |    **97.33**    |   **62.51**   |          95.01         |         95.35        |       62.06       |      70.86      |       98.87       |    **84.43**   |
>
> We also conduct experiments on CogVideoX. From the results below and the Figure 18 of the Appendix, SimpleGVR can generalize to CogVideoX.
>
> |           Method          |    MUSIQ    |   MANIQA   |  NIQE(↓)  |   CLIPIQA  | DOVER-Techinical | DOVER-Aesthetic | DOVER-Overall | Background Consistency | Subject  Consistency | Aesthetic Quality | Imaging Quality | Motion Smoothness | VBench-Average  |
> |:-------------------------:|:-----------:|:----------:|:---------:|:----------:|:----------------:|:---------------:|:-------------:|:----------------------:|:--------------------:|:-----------------:|:---------------:|:-----------------:|:---------------:|
> |       CogVideoX-5B-480$\times$720        |   46.3907   |   0.2350   |   6.201   |   0.3202   |       9.82       |      96.00      |     46.37     |          95.33         |         94.50        |       59.25       |      62.15      |       97.91       |      81.83      |
> |       CogVideoX-V1.5-768$\times$1360       |   53.4747   |   0.2575   |   4.893   |   0.3994   |       10.99      |      95.49      |     47.73     |          96.01         |         96.10        |       57.01       |      65.15      |       98.29       |      82.51      |
> | CogVideoX-5B-480$\times$720 + SimpleGVR | **66.0243** | **0.3413** | **4.064** | **0.5336** |     **16.16**    |    **97.33**    |     59.12     |          94.48         |         94.08        |       58.33       |      71.87      |       97.53       |    **83.23**    |
>
> ---
> `[W2] Flow-based degradation may be complex and error-prone due to multiple stages.`
>
> Yes, the flow-based degradation is a hand-crafted pipeline that may not be optimal. By observating the data characteristics of the base T2V model, the goal of flow-based degradation is to generate low-resolution videos that resemble the outputs of the base T2V model to a reasonable degree. As shown in Table 4 and Figure 11, the flow-based degradation strategy indeed improves the model’s performance.
>
> Also, it's important to note that our another model-guided degradation can synthesize the low-resolution videos that inherit the distribution of the base T2V model. The model-guided degradation strategy plays a complementary role to the flow-based degradation, and the ablation study in Table 4 (main paper) confirms the value of combining both strategies.

---

> ### Author Response · Authors · 2025-11-24
> **Official Comment by Authors (Part 2/2)**
>
> `[Q1]: The rationale behind the proposed low-res latent injection method (increase the channel/temporal, spatial interpolation, down the temporal/channel)? Ablation study of directly increasing the spatial dimension of low-res latents to match the high-res latents?`
>
> Note that our VAE invloves temporal compression. The latent upsampler is designed to roughly simulate an inverse VAE architerture, namely decoder + upscaling (alike in pixel domain) + encoder. Following the decoder of VAE, before performing spatial upscaling, the temporal dimension of the latent is first expanded so that each frame in the expanded latent corresponds to a single frame in RGB space. This design prevents inter-frame signal aliasing during the spatial upscaling process. Besides, channel expansion provides flexibility for feature transformation, as the same mechanism in VAE.
>
> As for the suggested ablation study, since the low-res and high-res latents share the same channel and temporal dimensions, spatial interpolation alone is sufficient to align the low-res latent with the high-res latent. This corresponds to Figure 2(b) in the main paper (interpolation + channel concatenation), with quantitative and qualitative results reported in Table 3 and Figure 10, respectively. Compared against our latent upsampler, directly latent-level interpolation damages the original fine struture and semantics in some degree (e.g., an extra ear appears in Figure 10(b)).
>
>
> `[Q2]: The details of the detail-aware sampler.`
>
> The detail-aware sampler establishes the sampling probability for each timestep during training by leveraging dedicated statistics that quantify detail variations at that particular timestep. Since the high-frequency details produced by SimpleGVR at different timesteps are based on a 50-step inference setting, we uniformly divide the training timestep range (0–1000) into 50 intervals, with each inference step corresponding to one interval. The absolute change of high-frequency details at each inference step is normalized to derive the sampling probability for the corresponding training interval, such that intervals with more significant high-frequency changes have higher sampling probabilities. Based on these probabilities, we construct a cumulative distribution function over the intervals. During training, for a selected interval, each point within the interval (20 points) is then sampled uniformly at random.
>
>
> `[Q3]: In interleaving temporal unit, how to resolve the memory consumption? Are all intermediate results cached?`
>
> In the implementation of the temporal interleaving unit, we concatenate multiple video chunks along the batch dimension and process them in parallel through the attention module. We do not specifically optimize for memory consumption, nor do we cache intermediate results. Caching intermediate activations is an excellent suggestion, and we will consider it in future work.
>
> [1] Chan, Kelvin CK, et al. "Investigating tradeoffs in real-world video super-resolution." Proceedings of the IEEE/CVF conference on computer vision and pattern recognition. 2022.
>
> [2] Wan, Team, et al. "Wan: Open and advanced large-scale video generative models." arXiv preprint arXiv:2503.20314 (2025).

---

> > ### Comment · Reviewer_tr3b · 2025-11-28
> >
> > Thanks authors for the response. It has addressed most of my concerns. Please include the clarification part in the final manuscript.
> >
> > One follow up question for the detail-aware sampler: to clarify, during training, you derive different sampling probability for different time steps, like higher noise might have higher trained probability. But the sampling steps during inference time are still uniformly selected like 1000, 980, 960, ..., 20, 0, right? If so, please clarify this in the final manuscript about what you did for both training and inference time.

---

> > > ### Author Response · Authors · 2025-11-28
> > > **Official Comment by Authors**
> > >
> > > We thank the reviewer for the positive feedback and for acknowledging our previous response.
> > >
> > > Regarding your follow-up question on the detail-aware sampler, we confirm that your understanding is correct: during training, we derive different sampling probabilities for different time steps (e.g., assigning higher probabilities to timesteps with higher noise or specific characteristics); during inference, the sampling steps remain uniformly selected (e.g., 1000, 980, 960, ..., 0) as in standard diffusion processes. We will explicitly clarify this in the final version of the manuscript as suggested.
> > >
> > > Thanks again for your constructive comments which have helped improve our paper.
> > >
> > > Best regards,
> > >
> > > The Authors of Submission 5131

---

### Author Response · Authors · 2025-11-24
**Common Response**

`[R1,R2,R3] The technique novelty and importance of our proposed latent upsampler.`

* **Its goal is non-trivial to achieve:** The goal of latent upsampler is to obtain a higher-resolution latent from a low-resolution latent with the represented information free of damage. We find that for VAE latent that involves temporal compression (which is a general case), conducting spatial interpolation neither on the latent directly nor some higher dimentional feature space via 3D CNN can not avoid distortion of fine structure and semantics. Please refere to the comparison presented in Fig.10 and Tab. 3 in the main paper.

* **Its archietecture offers unique insight:** Our key idea draws inspiration from a heavy baseline architecture comprising a VAE decoder, per-frame pixel interpolation, and a VAE encoder. A critical insight here is the necessity to temporally expand the latent representation such that each latent frame corresponds directly to a single frame of the original video. This explicit correspondence then enables effective per-frame spatial interpolation.
Considering computational efficiency, we employ a relatively compact network (specifically, four 3D ResBlocks) to achieve the required temporal expansion and subsequent compression. Our experiments have robustly confirmed the effectiveness of this design.

---

### Author Response · Authors · 2025-11-24
**Global Reply**

Dear Reviewers,

We sincerely thank all reviewers for their thoughtful comments and constructive feedback.

We have carefully considered each point and provided clarifications and justifications accordingly. Detailed responses are included below. In addition, we have uploaded a **revised PDF** in which all modifications are **highlighted in blue**. Specifically, we have made the following updates:

* **Section 3.2, Figure 10, Table 3**: Added more analysis of the latent upsampler.
* **Table2**: Added results for additional recent methods and evaluation metrics.
* **Appendix A.4**: Added a more comprehensive user study.
* **Appendix A.8**: Added the details of the AIGC100.
* **Appendix A.9**: Added the performance of SimpleGVR on VideoLQ.
* **Appendix A.10, Table 7, Table 8, Figure 17, Figure 18**: Added the performance of SimpleGVR on other T2V models (Wan, CogVideoX).
* **Figure 8**: Added more explanation for Figure 8.


We hope that our explanations adequately address all concerns. Please feel free to let us know if any further details or clarifications would be helpful.

Best regards,
Authors of Paper #5131

---

### Author Response · Authors · 2025-12-03
**Rebuttal Summary for Submission 5131**

Dear Area Chair,

Thank you for taking over the evaluation of our submission. We provide a brief summary of the rebuttal. In the previous round, reviewers generally acknowledged the value of our work and the validity of our experiments. For the Area Chair’s convenience, we summarize the main concerns and how they were addressed.

---
### **Key Concerns Raised by Reviewers and Our Responses**

1. **Common technical clarification: Novelty and importance of latent upsampler**
This clarification addresses questions regarding the technical novelty and necessity of our latent upsampler design raised by reviewers tr3b, D3ZN, and ExS4. **In the common response**, we emphasize the goal of latent upsampler is non-trivial to achieve and the design of latent upsampler is motivated by unique obervation that can achieve the goal effectively but multiple baselines failed.

1. **tr3b (6, keeps positive score)**
   * **Generalization to other T2V models.** We first clarified that such inference-time genenralization is beyond role scope of our work. Anyway, as an extended study, we also added the experiments for generalization evaluation on **Wan-14B** and **CogVideoX**, showing effective performance (**Appendix A.10, Table 7, 8, Fig. 17, 18**).
   * **Error accumulation of flow-based degradation.** We acknowledged it is a hand-crafted pipeline and may not be optimal, but experiments confirms its benefits (**Table 4**). Besides, it is just part of our degradation simulation, complemented by another model-guided strategy.
   * **Justification to the temporal expansion design of Latent Upsampler.** We explained that the "temporal expansion $\to$ interpolation" design simulates an inverse VAE process to prevent inter-frame aliasing, which direct interpolation fails to address (verified by **Fig. 10(b)**) (**Sec. 3.2**).
2. **D3ZN (6, no post-rebuttal response)**
   * **Motivation Illustration of degradation simulation.** To make the motivation more obvious, we updated **Fig. 4** in the main paper.
   * **Evaluation only on datasets generated by the chosen T2V model.** This is determined by our problem setting that the SimpleGVR is only required to cooperate with the chosen T2V base model for video super resolution, rather than for general VSR task.
   * **Limited baselines and metrics.** We added comparisons with **5+ SOTA methods** (MGLD, DLoRAL, Dove, DiffVSR, and SeedVR2) and included metrics like **CLIPIQA, MANIQA**, and **warp error** (**Table 2, Sec. 4.2**).
3. **ExS4 (4 $\to$ 6)**
   * **Originality w.r.t existing works.** Our unique contribution lies in two aspects: (i) we perform vsr in the latent space directly without the VAE decoding, pixel-space resizing, and re-encoding process like existing works. This is enabled by a novel design of latent upsampler. (ii) we conduct a thorough analysis over the degradation simulation that ensures the distribution alignment with the chosen T2V model.
   * **Fairness of experiment settings.** Our objective is to achieve computational decoupling for high-resolution Text-to-Video (T2V) generation, which are fully different with those existing VSR models that are generally designed for general VSR tasks. This fundamental difference in problem settings makes a completely fair comparison challenging. Therefore, our comparison with these existing publicly available general VSR models is intended only to justify the need for our proposed approach, rather than to directly benchmark them under fully aligned settings.
   * **Generalization to other T2V models.** We added results on **Wan-14B**, **CogVideoX** (**Appx. A.10**) to demonstrate robustness.
   * **Comparison with strong baselines.** We incorporated **DiffVSR**, **MGLD**, **DLORAL**, **DOVE**, and **SeedVR2**, demonstrating SimpleGVR's superiority (**Table 2**).
   * **Subjective evaluation.** We conducted a comprehensive **user study** (**Appx. A.4**) and added temporal consistency metrics ($E_{warp}$), where SimpleGVR showed strong preference (**Table 2**).

4. **w3HT (6, no post-rebuttal response)**
   * **Cascade pipeline is not new.** Our unique contribution is not the cascade pipeline. Instead, three aspects of non-trivial exploration are made in this setting. Please check our rebuttal for details.
   * **Temporal consistency is not well evaluated.** We added a **user study** on temporal consistency and included the quantitative **warp error** metric (**Appx. A.4, Table 2**).
   * **A user study is recommended.** We conducted a comprehensive **user study** comparing all baselines, showing SimpleGVR is the most preferred approach (**Appx. A.4**).


---
### **Conclusion**
Our rebuttal provides detailed response to the reviewers’ concerns. All updates in the revised manuscript are highlighted in **blue**.

We hope these revisions clearly convey SimpleGVR’s contributions and benefits to the ICLR community, and we appreciate your time in evaluating our work.

---

Best regards,
*Submission5131 Authors*

---

### Meta-Review · Area_Chair_iPN4 · 2025-12-17

**Summary:**

The reviewers generally concern about the 1) novelty, 2) evolution, 3) generalizability, and 4) complexity of the method.

1. Novelty: the reviewers raise concerns about the cascaded pipelines and latent upsampler.

2. Evaluation: The main concerns are missing baselines and limited metrics.

3. Generalizability: The reviewers question the dependency on the base model and dataset bias.

4. Complexity: The reviewers find that the motivation of flow-based degradation is unclear, and the pipeline is complex.

**Reviewer Concerns:**

Most of the concerns, including generalizability, baselines, and novelty, are addressed in the rebuttal, and acknowledged by the reviewers. However, there are remaining concerns, including the validity of flow-based degradation and dataset bias.

**Reviewer Scores:**

This paper receives initial ratings of (4, 6, 6, 6) and reviewer ExS4 then increases the score to 4. Reviewer tr3b mentions that most of the concerns are addressed but do not explicitly mention that score will be raised.

Based on the review, rebuttal, and discussion, the AC recommends an acceptance.

---

### Decision · Program_Chairs · 2026-01-26

Accept (Poster)